# CACHE-TO-CACHE: DIRECT SEMANTIC COMMUNICATION BETWEEN LARGE LANGUAGE MODELS

**Tianyu Fu**[*1,2], **Zihan Min**[*1], **Hanling Zhang**[*3], **Jichao Yan**[1],
**Guohao Dai**[4,2], **Wanli Ouyang**[3,5,6], **Yu Wang**[†1]

[1]Tsinghua University    [2]Infinigence AI    [3]The Chinese University of Hong Kong
[4]Shanghai Jiao Tong University    [5]SLAI    [6]Shanghai AI Laboratory

## ABSTRACT

Multi-LLM systems harness the complementary strengths of diverse Large Language Models, achieving performance and efficiency gains that are not attainable by a single model. In existing designs, LLMs communicate through text, forcing internal representations to be transformed into output token sequences. This process both loses rich semantic information and incurs token-by-token generation latency. Motivated by these limitations, we ask: *Can LLMs communicate beyond text?* Oracle experiments show that enriching the KV-Cache semantics can improve response quality without increasing cache size, supporting KV-Cache as an effective medium for inter-model communication. Thus, we propose Cache-to-Cache (C2C), a new paradigm for direct semantic communication between LLMs. C2C uses a neural network to project and fuse the source model's KV-cache with that of the target model to enable direct semantic transfer. A learnable gating mechanism selects the target layers that benefit from cache communication. Compared with text communication, C2C utilizes the deep, specialized semantics from both models, while avoiding explicit intermediate text generation. Experiments show that C2C achieves 6.4-14.2% higher average accuracy than individual models. It further outperforms the text communication paradigm by approximately 3.1-5.4%, while delivering an average 2.5× speedup in latency. Our code is available at `https://github.com/thu-nics/C2C`.

## 1 INTRODUCTION

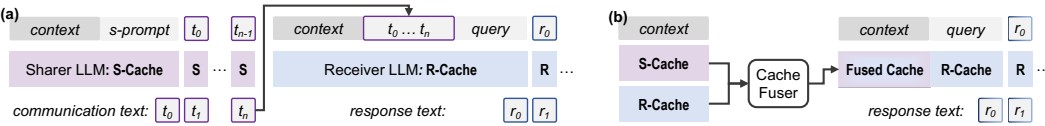

Figure 1: (a) Previous Text-to-Text (T2T) communication passes information through explicit text generation. (b) Our Cache-to-Cache (C2C) communication directly projects and merges KV-Cache with rich semantics from different LLMs.

With the rapid progress of Large Language Models (LLMs) (Guo et al., 2025; Yang et al., 2025a; OpenAI, 2025), they are now applied across increasingly diverse domains and tasks. To meet versatile demands, LLMs are trained with distinct focuses, such as coding (Hui et al., 2024), mathematics (Yang et al., 2024a), visual understanding (Bai et al., 2025), edge computing (Zhang et al., 2024b), and so on. Meanwhile, general-purpose LLMs can also simulate specialized capabilities through prompt engineering, enabling flexible role adaptation across downstream applications.

Leveraging the diversity of LLMs, many multi-LLM systems are proposed to further enhance overall performance and efficiency (Guo et al., 2024; Tran et al., 2025). In **collaborative multi-LLM**

---

[*]Equal contribution.
[†]Corresponding author: Yu Wang (yu-wang@tsinghua.edu.cn).

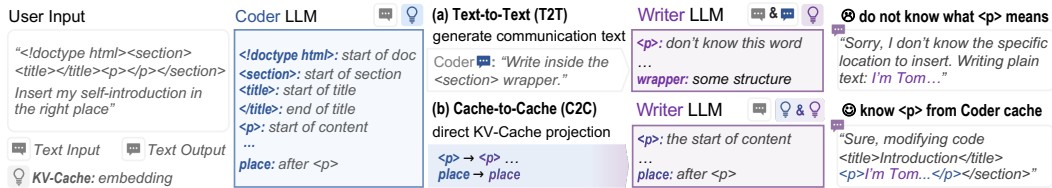

Figure 2: Conceptual comparison of T2T and C2C communication in a Coder-Writer collaboration example. In T2T, the Coder's ambiguous text instruction fails to convey the structural semantics of <p> as a paragraph separator, causing the Writer to misplace the content. C2C directly projects the Coder's KV-Cache into the Writer, transferring both the semantic understanding and precise insertion location without intermediate text generation.

**systems** (Li et al., 2023; Wu et al., 2023), LLMs are assigned distinct roles and proactively exchange text messages. Mirroring human collaboration, these systems accumulate partial understandings or sub-solutions from different agents via verbal communication. They harness the collective capabilities of multiple LLMs to solve complex problems that a single model cannot. By contrast, **routing-based** multi-LLM inference systems rely on passive context inheritance rather than active message exchange. These systems coordinate models of varying parameter sizes or reasoning depths for more dynamic and efficient responses (Li et al., 2024; Fu et al., 2025a; Ong et al., 2024; OpenAI, 2025). Downstream models inherit the context from preceding models in multi-round conversations, then generate follow-up responses to the new questions based on their own understanding of the conversation history.

However, current text-to-text (T2T) interfaces restrict information exchange among LLMs, particularly when conveying rich or diverse semantic interpretations of a shared context. As illustrated in Figure 2, these limitations arise from several inherent constraints of T2T communication. First, as a low-bandwidth medium, text introduces an information bottleneck. The high-dimensional internal representations must be repeatedly compressed into linear strings and then decompressed by the receiver LLM. When models differ in knowledge or assigned roles, some signals may be irrecoverable (e.g., interpreting <p> as a section marker). Second, natural language is inherently ambiguous, with idioms, underspecified references, and vague expressions. Although recent agent protocols aim to standardize text messages (Anthropic, 2024; Surapaneni et al., 2025), rigid templates remain insufficient for flexible, open-domain collaboration. Third, T2T communication incurs noticeable latency. Every exchange requires exhaustive, token-by-token decoding of contextual explanations in sequence. These limitations motivate a key question:

*Can LLMs communicate beyond text?*

In this work, we explore using KV-Cache as the medium for LLM communication. KV-Cache is a naturally richer representation than text. It also enables fully parallel communication through direct projection, avoiding the slow sequential decoding in text exchanges. Our oracle experiments show that (1) enriching KV-Cache under the same context length increases accuracy, (2) KV-Cache is convertible between LLMs, (3) different LLMs encode distinct semantic understandings and contextual knowledge of the same input, reflecting their complementary strengths.

Encouraged by these findings, we propose Cache-to-Cache (C2C), a new paradigm for richer and faster multi-LLM communication. As shown in Figure 1(b), C2C projects the KV-Cache from a source model into the space of a target model and merges them through a neural cache fuser. Experiments show that C2C achieves 6.4-14.2% higher average accuracy than individual models. It further outperforms the T2T paradigm by approximately 3.1-5.4%, while delivering an average 2.5× speedup in latency.

## 2 RELATED WORK

### 2.1 KV-CACHE SHARING AND REUSE

Based on the similarity of KV-Cache between layers, intra-model cache sharing methods (Yang et al., 2024b; Wu & Tu, 2024; Sun et al., 2024; Brandon et al., 2024; Wu et al., 2025) have been proposed to reuse shallow layers' KV-Cache for deeper layers to accelerate single LLM inference.

Another research focus is to reuse a portion of KV-Cache (e.g., common prefix, reference documents) for the same model in multiple user queries (Bang, 2023; Ye et al., 2024; Yao et al., 2024; Qin et al., 2024; Yang et al., 2025b; Ye et al., 2025; Eyuboglu et al., 2025). DroidSpeak (Liu et al., 2024a) extends cache reuse to models fine-tuned from the same base model. Unlike existing works that focus on computational efficiency through cache reuse, our approach leverages the KV-Cache as a medium for semantic transfer between LLMs. Furthermore, unlike existing cache sharing methods that are restricted to only a single model or models with identical structure and size, our method supports sharing across different model families and varying model sizes.

## 2.2 MULTI-LLM SYSTEMS

**Collaborative multi-LLM systems**. Collaborative systems treat multiple LLMs as peers that exchange information to improve collective performance. Chain-of-Agents (Zhang et al., 2024c) and MetaGPT (Hong et al., 2023) create sequential message flows where agents directly communicate using natural language interfaces. Mixture-of-Agents (Wang et al., 2024), DyLAN (Liu et al., 2024b), and Owl (Hu et al., 2025) introduce layered communication architectures. Target LLMs aggregate messages from multiple models using voting or summarization mechanisms. Multi-agent debate methods (Estornell & Liu, 2024; Liang et al., 2024; Du et al., 2023) involve iterative communication rounds, letting LLM agents discuss and refine responses. Recent works such as MCP Anthropic (2024) and A2A Surapaneni et al. (2025) establish formal text protocols beyond natural language, standardizing agent interaction and tool usage in collaborative multi-LLM systems. These approaches rely on text-level interfaces, where communication requires one model to generate text token-by-token and another to ingest it as input. Our work explores a deeper and more efficient collaboration (Pidkuiko & Starkov, 2025; Zheng et al., 2025b) by directly sharing internal KV-Cache representations.

**Routing-based multi-LLM inference systems**. To accelerate LLM inference, several systems leverage multiple models with different capabilities and costs. Dynamic model selection methods (OpenAI, 2025; Ong et al., 2024; Feng et al., 2024; Ning et al., 2024) route queries to different models with varying sizes and configurations to balance efficiency and performance. Token-level routing methods (Zhang et al., 2024a; Shen et al., 2024; Zheng et al., 2025a; Fu et al., 2025a) enable finer-grained selection, utilizing smaller models for simple token generation within the reasoning process of complex tasks. While these systems achieve efficiency through strategic model switching, they either completely drop context from other models, or simply rely on their own understandings of the context. Without understanding sharing, smaller models cannot benefit from the richer representations already computed by larger models.

## 3 METHOD

### 3.1 PRELIMINARIES

**LLM inference**. Autoregressive LLM inference involves two stages: *prefill* and *decode*. Prefill encodes the full input to produce the first output token; decode then generates subsequent tokens iteratively using the last token and the cached key–value (KV) states. Formally, let $X_{[0:n]} = [x_0, \ldots, x_{n-1}]$ be the input token sequence. After prefill, LLM produces a per-token KV-Cache $\mathcal{C}(X_{[0:n]}) = [c_0, \ldots, c_{n-1}] \in \mathbb{R}^{n \times d}$. For notation brevity, $d$ denotes the KV dimensionality that is flattened from all layers into a single vector per token. The range subscripts are omitted when clear. During decoding, with the current token $y_i$ and caches from the input and the generated prefix, the next token is predicted as

$$y_{i+1} = \mathcal{P}\big(y_i \mid \mathcal{C}(X) \oplus \mathcal{C}(Y_{[0:i]})\big), \tag{1}$$

where $\oplus$ denotes sequence-wise concatenation. The cache updates as $\mathcal{C}(Y_{[0:i+1]}) = \mathcal{C}(Y_{[0:i]}) \oplus \mathcal{C}(y_i)$.

**LLM communication**. In LLM communication scenarios, we define the LLM that provides contextual understanding or knowledge as *Sharer*, and the one that utilizes it as *Receiver*.

### 3.2 ORACLES FOR CACHE-TO-CACHE COMMUNICATION

We aim to explore whether LLMs can have direct semantic communication through KV-Cache. Specifically, we design two oracle experiments to answer the following questions: (1) *Benefit*: can

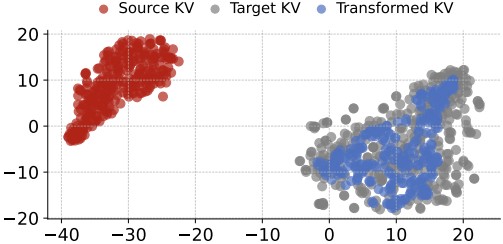

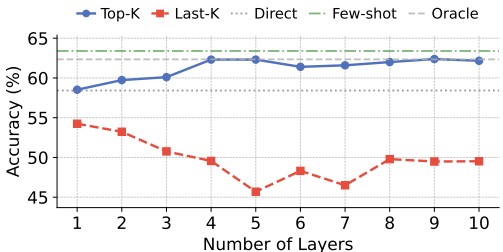

Figure 3: T-SNE visualization of KV-Cache representations from the source (Qwen3-4B), target (Qwen3-0.6B), and the transformed cache. After transformation, the source cache falls within the target's representation space.

Figure 4: Effect of selectively enriching different numbers of layers' KV-Cache on accuracy. Enriching more best-performing layers yields increased accuracy, while enriching the worst-performing ones declines accuracy.

| Method | Cache Len. | Cache Enrich. | Acc. (%) |
|---|---|---|---|
| Direct | $|X|$ | No | 58.42 |
| Few-shot | $|E| + |X|$ | Yes | 63.39 |
| Oracle | $|X|$ | Yes | 62.34 |

Table 1: Cache enrichment experiment. *Oracle* prefills on exemplars ($E$) and question ($X$), then drops the exemplar cache before decoding, isolating semantic enrichment from cache length.

| | | Average Effective Rank | |
|---|---|---|---|
| **Type** | **Sharer** | **Receiver** | **C2C** |
| K Cache | 539 | 388 | 395 |
| V Cache | 689 | 532 | 560 |

Table 2: Average effective rank of KV-Cache from Sharer, Receiver, and the C2C-fused one. The increase after fusion indicates that C2C enriches the Receiver KV-Cache semantics.

a model's capabilities be improved through KV-Cache semantic enrichment without extending sequence length? (2) *Convertibility*: can the KV-Cache of one model be effectively utilized by another model?

### 3.2.1 CACHE ENRICHMENT ORACLE

To validate the benefit of cache enrichment, we first explore whether the semantic quality of KV-Cache can be improved without increasing its size. Few-shot prompting suggests this might work: providing *exemplars* $E$ before the *question* $X$ often improves accuracy. But does this arise from attending to more context tokens, or from $E$ enriching how $X$ is embedded in KV-Cache?

We evaluate this via three setups: (1) *Direct*: prefill on $X$ only and decode with $\mathcal{C}(X)$; (2) *Few-shot*: prefill on $E \oplus X$ and decode with $\mathcal{C}(E \oplus X)$ (longer cache); (3) *Oracle*: prefill on $E \oplus X$ but *discard* the exemplar segment and keep only the question-aligned slice

$$\mathcal{C}^*(X) \ = \ \mathcal{C}_{[\,|E|:|E|+|X|\,]}(E \oplus X), \tag{2}$$

so that decoding uses a question-length cache with no extra tokens. Here, $|\cdot|$ denotes sequence length. In Equation 1, this corresponds to substituting $\mathcal{C}(X)$ with $\mathcal{C}^*(X)$ before decoding.

Comparing *Direct* and *Oracle* isolates the effect of cache enrichment: any gain arises from the richer question embeddings induced by $E$, not from attending to additional token caches as in *Few-shot*. As shown in Table 1, the *Oracle* setup improves response quality at the same cache length.

Additionally, we analyze how cache enrichment affects different transformer layers. Our findings show substantial variation across layers: while some layers benefit from cache enrichment, others experience performance degradation (details in Appendix A.2.1). Furthermore, these layer-wise effects accumulate as more layers are augmented. As shown in Figure 4, selectively applying cache enrichment to the top-performing layers (e.g., top-5) yields slightly higher accuracy than enriching all layers, while targeting the worst-performing layers leads to accuracy decline. This finding guides the gating mechanism of our cache fuser (Section 3.3.2).

Figure 5: Cache fuser architecture and training scheme. *Fuser* projects and combines KV-Caches, then adds the result to Receiver's cache through a learnable gate. LLMs are frozen during training.

### 3.2.2 CACHE TRANSFORMATION ORACLE

To verify that one model's KV-Cache can be utilized by another, we conducted cross-model transformation experiments. We train a 3-layer MLP to map the KV-Cache from a source LLM (Qwen3-4B) to a target LLM (Qwen3-0.6B), with more setups detailed in Appendix A.3.2.

T-SNE visualizations in Figure 3 reveal that the raw KV-Caches of the two LLMs are far apart in representation space. After transformation, the mapped KV-Cache lies inside the target model's representation space. These results demonstrate that KV-Caches from different models are generally convertible, as the transformed cache is covered by the target model's representation space.

One thing to note is that the transformed cache occupies only a smaller subset of the target's space. It indicates that the source model's semantic information cannot fully cover the target's, despite the source being larger. This reflects inherent differences in how each model encodes context. Another observation also supports this interpretation: the correct-answer sets of different models exhibit limited overlap (Figure 7), despite the comparable aggregated accuracy of respective models. These findings suggest that if specialized contextual understanding from different models can be successfully projected and fused, it may harness the complementary strengths of the respective models.

## 3.3 C2C DESIGN

### 3.3.1 OVERVIEW

Building on the oracle experiments, we propose the C2C scheme. Its core objective is to extract useful contextual understanding or knowledge from one model (the Sharer) and fuse it into another model (the Receiver).

In general, the C2C paradigm contains a set of key/value cache fusers $\mathcal{F}$ and a layer mapping strategy $\mathcal{G}$. During the prefill stage, fuser $\mathcal{F}_n$ takes the $n$th layer cache of the Receiver Model $\mathcal{C}_n(X)$ and the corresponding $\mathcal{G}(n)$th layer cache of the Sharer Model $\mathcal{C}^{\mathcal{S}}_{\mathcal{G}(n)}(X)$ and generates the corresponding fused cache with residual connection:

$$\mathcal{C}^{\mathcal{F}} = \left\{ \mathcal{C}_n(X) + \mathcal{F}_n\Big(\mathcal{C}_n(X),\ \mathcal{C}^{\mathcal{S}}_{\mathcal{G}(n)}(X)\Big) \right\}_{n=1}^{N} \tag{3}$$

During decoding, with the current token $y_i$ and caches from the input and the generated prefix, the next token is predicted as:

$$y_{i+1} = \mathcal{P}\left( y_i \big| \mathcal{C}^{\mathcal{F}}(X) \oplus \mathcal{C}(Y_{[0:i]}) \right) \tag{4}$$

### 3.3.2 FUSER STRUCTURE

To enhance the Receiver's KV-Cache without destructive overwriting of its information, the fuser is designed under a residual integration principle. As shown in Figure 5, it contains three key modules:

(1) **Projection module** concatenates the Receiver's KV-Cache with the Sharer's KV-Cache, then processes the concatenated features through a projection layer followed by a feature fusion layer.

(2) **Dynamic weighting module** applies an input-aware head modulation layer to dynamically reweight the projected information.

(3) **Learnable gate** introduces a trainable per-layer gate value that decides whether to inject the Sharer's context. The gate applies a Gumbel-sigmoid with temperature annealing to smoothly transition from differentiable during training to binary at inference.

We also explore a more complex yet potentially more powerful fuser variant in Appendix A.1.3.

### 3.3.3 MODEL ALIGNMENT

Fusing KV-Caches across model families and sizes requires alignment at two levels: tokens and layers. For *token alignment*, different tokenizers may produce slightly varied token sequences for the same input. We align them by decoding each Receiver token into its string form and re-encoding it using the Sharer's tokenizer. When one-to-many mappings occasionally occur, we select the Sharer token with maximal string coverage to preserve information. For *layer* alignment, we adopt a terminal alignment strategy: the final layers of both models are aligned first, then the penultimate layers, and so on in reverse order until reaching the shallower model's first layer. Details in Appendices A.1.1 and A.1.2.

### 3.3.4 TRAINING SCHEME

During training, we freeze both the Sharer and Receiver models, training only the C2C module for KV-Cache fusion. We employ standard next-token prediction loss on the Receiver's response predictions, similar to supervised fine-tuning (SFT). The key difference is that the Receiver predicts responses conditioned on fused KV-Cache rather than its own.

The training procedure consists of three stages: (1) Forward: both models encode the input context to produce their respective KV-Caches. (2) Fusion: the C2C module fuses both KV-Caches and replaces the Receiver's cache. (3) Supervision: the Receiver prefills the response using the fused cache, and gradients backpropagate through C2C to minimize prediction loss.

## 4 EXPERIMENT

### 4.1 SETUP

We brief the experiment setups here, with more details in Appendix A.3.

**Models**. We evaluate C2C across various model families, including Qwen2.5 (Yang et al., 2024a; Hui et al., 2024), Qwen3 (Yang et al., 2025a), Llama3.2 (Dubey et al., 2024), and Gemma3 (Team et al., 2025). To test generalizability, we select different configurations for the Sharer-Receiver model combinations, including models of different generations (Qwen3 and Qwen2.5), different families (Qwen, Llama, and Gemma), different sizes (0.6B to 14B), different specializations (general, code, and math model), and different training stages (pretrained and instruction fine-tuned models). For ablative and diagnostic analyses (scaling behavior, ablation study, behavior analysis), we fix the Receiver and Sharer to Qwen3 models unless otherwise specified. This consistency eliminates confounders from model alignment and isolates the core impact of C2C.

**Baselines**. We compare C2C with two LLM collaboration methods to contextualize performance: (1) Text-to-Text (T2T) communication: models collaborate through an analyze-then-respond hand-off for each query. The Sharer generates an analytical text of key information to solve the input question. This text is concatenated with the original question and fed to the Receiver to mirror standard collaborative pipelines. Corresponding prompts are in Appendix A.3.6. (2) Query-level routing (Ong et al., 2024): models collaborate by selecting the appropriate LLM for different queries. We also include individual model performance (Sharer or Receiver alone) to establish a lower bound for collaborative gains.

**Benchmarks**. We evaluate on four widely used benchmarks spanning reasoning, knowledge, and language domains to ensure comprehensive coverage. OpenBookQA (Mihaylov et al., 2018) for fact-based reasoning, MMLU-Redux (Gema et al., 2025) for knowledge in the general domain, ARC-Challenge (ARC-C) (Clark et al., 2018) for scientific and logistic reasoning, and C-Eval (Huang et al., 2023) for comprehensive knowledge in the Chinese domain.

Table 3: Average token count and inference time breakdown on MMLU-Redux (Sharer: Qwen2.5-0.5B-Instruct, Receiver: Qwen3-0.6B). *Includes 90ms KV-Cache fusion time.

| Metric | Receiver-only | Sharer-only | Text-to-Text | | Cache-to-Cache | |
|---|---|---|---|---|---|---|
| | | | Sharer | Receiver | Sharer | Receiver |
| Input Tokens | 170 | 187 | 103 | 332 | 170 | 170 |
| Output Tokens | 11 | 19 | 80 | 10 | 0 | 12 |
| Prefill Time (ms) | 27 | 20 | 21 | 32 | 20 + 90* | 27 |
| Decode Time (ms) | 281 | 326 | 1312 | 231 | 0 | 308 |
| Total Time (ms) | 308 | 346 | 1596 | | 445 | |

**Training dataset**. To ensure the generalizability of C2C, we utilize the first 500k samples of the OpenHermes2.5 Dataset (Teknium, 2023), a general finetuning dataset, to train C2C fusers. To reduce training cost, we use MMLU as the training set for scaling behavior and behavior analysis experiments, unless otherwise specified.

**Evaluation settings**. We use average accuracy as the performance metric. We use text generation and answer extraction as the evaluation mode for C2C and baselines, with the max generation length set to 64 for multi-choice benchmarks. All experiments are conducted in the zero-shot setting with zero generation temperature to ensure reproducibility. We use average inference time as the efficiency metric, measured using a single NVIDIA A100 GPU with batch size one.

## 4.2 MAIN RESULTS

**Performance**. As shown in Table 4, C2C consistently improves the Receiver model performance across different settings and benchmarks. Representative example output is provided in Appendix A.4.4. After applying C2C, accuracy increases by an average of 11.00%, 9.64%, and 11.88% across the three Sharers. Compared with text-to-text communication, C2C achieves an average accuracy increase of 5.36%, 4.15%, and 3.06%. Query-level routing prioritizes efficiency but limits accuracy to the better of the two original models. Notably, Qwen3-4B Base as the Sharer often ignores instructions, resulting in poor standalone performance and excessively long T2T communication times. In contrast, C2C bypasses this issue, highlighting an interesting use case in which a weaker instruction-tuned Receiver can leverage a stronger base model's knowledge via C2C, even when base model cannot follow instructions. We also explore strong-to-weak communication (details in Appendix A.2.2) using Qwen3-4B as the Sharer, showing that C2C effectively enables the Receiver to benefit from a stronger Sharer.

**Efficiency**. As shown in Table 4, C2C achieves significant speedups of 3.46×, 1.51×, and 14.41× over T2T by eliminating intermediate text generation. As detailed in Table 3, T2T requires the Sharer to decode 80 output tokens, incurring 1312ms of decoding overhead, whereas C2C replaces this sequential decoding with parallel cache fusion in 90ms. Further analysis of Llama3.2-1B's exceptionally fast inference is provided in Appendix A.4.3.

## 4.3 SCALING BEHAVIOR

**Scaling sequence lengths**. We evaluate how C2C scales with respect to sequence length on long-context tasks from the LongBenchV1 benchmark. All C2C fusers are trained and tested on different sets of LongBenchV1. As shown in Table 5, C2C consistently outperforms text-to-text communication across all sequence-length intervals. This demonstrates C2C's advantages across input length ranges. More detailed setups and results are in Appendices A.2.2 and A.3.4.

**Scaling model sizes**. We investigate how C2C scales with respect to the Sharer and Receiver model sizes. All C2C fusers are trained on MMLU's auxiliary train split and evaluated on MMLU-Redux. As shown in Figure 6, the $x$-axis denotes Sharer size (Qwen2.5-Instruct series), the $y$-axis shows accuracy gains of C2C over Receiver-only baselines ($\Delta$ Accuracy), and each curve represents a Receiver from the Qwen3 series. We find that the accuracy improvements of C2C generally increase faster than T2T. This trend shows that when the Sharer possesses richer knowledge, C2C is able to

Table 4: Accuracy (%) and inference time (seconds) of different communication methods across four benchmarks. The Receiver is fixed as Qwen3-0.6B, paired with three different Sharers.

| Sharer | Task | Metric | Receiver | Sharer | Routing | Text-to-Text | Cache-to-Cache |
|---|---|---|---|---|---|---|---|
| Qwen2.5-0.5B | MMLU-Redux | Acc | 35.53 | 38.42 | 35.58 | 41.03 | **42.92** |
| | | Time | 0.29 | 0.34 | 0.27 | 1.52 | 0.40 |
| | OpenBook | Acc | 39.20 | 45.60 | 40.80 | 44.00 | **52.60** |
| | | Time | 0.27 | 0.35 | 0.29 | 0.81 | 0.30 |
| | ARC-C | Acc | 41.04 | 42.09 | 40.70 | 49.48 | **54.52** |
| | | Time | 0.29 | 0.39 | 0.29 | 1.00 | 0.36 |
| | C-Eval | Acc | 32.04 | 40.21 | 34.61 | 35.88 | **41.77** |
| | | Time | 0.26 | 0.31 | 0.26 | 1.51 | 0.34 |
| Llama3.2-1B | MMLU-Redux | Acc | 35.53 | 32.30 | 33.38 | 43.32 | **44.42** |
| | | Time | 0.29 | 0.06 | 0.18 | 0.75 | 0.50 |
| | OpenBook | Acc | 39.20 | 32.60 | 36.40 | 41.20 | **47.80** |
| | | Time | 0.26 | 0.07 | 0.17 | 0.70 | 0.43 |
| | ARC-C | Acc | 41.04 | 33.57 | 37.22 | 50.00 | **53.39** |
| | | Time | 0.28 | 0.07 | 0.18 | 0.70 | 0.47 |
| | C-Eval | Acc | 32.04 | 31.31 | 31.92 | 35.27 | **40.77** |
| | | Time | 0.25 | 0.04 | 0.15 | 0.71 | 0.49 |
| Qwen3-4B-Base | MMLU-Redux | Acc | 35.53 | 1.03 | 16.39 | 43.87 | **43.95** |
| | | Time | 0.29 | 2.06 | 0.28 | 7.54 | 0.45 |
| | OpenBook | Acc | 39.20 | 2.20 | 22.20 | 46.40 | **53.20** |
| | | Time | 0.26 | 1.98 | 0.27 | 5.08 | 0.34 |
| | ARC-C | Acc | 41.04 | 1.48 | 19.65 | 53.91 | **55.39** |
| | | Time | 0.28 | 2.06 | 0.28 | 6.56 | 0.40 |
| | C-Eval | Acc | 32.04 | 5.65 | 15.10 | 38.92 | **42.79** |
| | | Time | 0.25 | 2.02 | 0.26 | 3.59 | 0.39 |

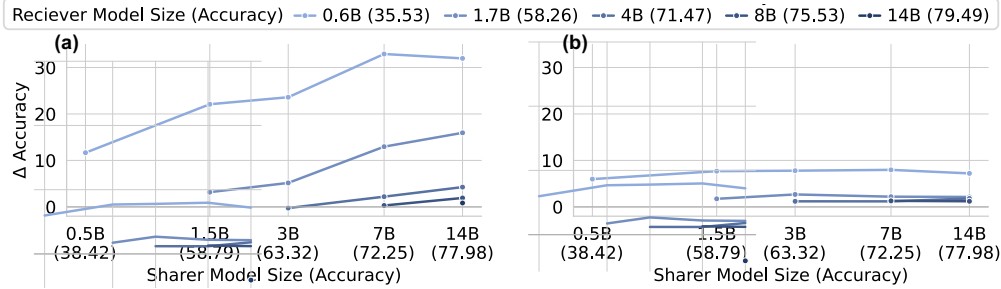

Figure 6: Accuracy improvements ($\Delta$Accuracy) on the MMLU-Redux benchmark. (a) C2C communication. (b) T2T communication. The $x$-axis denotes the Sharer model from the Qwen2.5-Instruct series, while the curves correspond to Receiver models from the Qwen3 series. The accuracy improvements of C2C generally increase faster than T2T.

more effectively transmit useful information to the Receiver. Note that the relative gains for larger Receivers are less pronounced due to their stronger baselines and higher overlap with the Sharer's knowledge.

**Different model combinations**. We test different Sharer-Receiver combinations, including different model families and different task-specific models. The result in Table 7 shows that C2C outperforms text-to-text communication on all five combinations by an average of 8.59% on MMLU-Redux. This supports that by employing C2C, the Receiver model can effectively utilize contextual understanding from different models to enhance performance. Notably, when using Qwen2.5-Math as the Sharer, the communication text becomes substantially longer, as analyzed in Appendix A.4.3. To further

| Length | Receiver | Sharer | T2T | C2C |
|--------|----------|--------|------|-------|
| 0-4k | 30.52 | 24.94 | 33.46 | **37.31** |
| 4-8k | 26.03 | 23.18 | 29.70 | **34.01** |
| 8k+ | 25.99 | 16.44 | 25.64 | **30.72** |

Table 5: Performance scaling with sequence length on LongBenchV1, using Qwen3-0.6B Receiver and Qwen2.5-0.5B Sharer.

| Setting | #Param. | OpenBook | ARC-C | MMLU | C-Eval |
|---------|---------|----------|-------|-------|--------|
| Single | 596M | 45.80 | 47.65 | 36.81 | 35.81 |
| Identical | 529M | 50.60 | 52.52 | 42.17 | 40.34 |
| C2C | 478M | **52.60** | **54.52** | **42.92** | **41.77** |

Table 6: Ablation of C2C improvement sources. *Single*: fine-tunes Receiver without any Sharer. *Identical*: C2C with the same LLM as both Sharer and Receiver. *C2C*: use different LLMs as Sharer and Receiver.

test the generalizability of C2C, we swap the Sharer and Receiver models. The results show that C2C robustly brings a 5.05% increase in accuracy while applying T2T results in a 6.30% decrease in performance.

Together, these experiments support the scalability of C2C as an effective and efficient new LLM communication paradigm.

| Pair Type | Receiver | Sharer | Metric | Receiver | Sharer | T2T | C2C |
|-----------|----------|--------|--------|----------|--------|------|-------|
| Heterogeneous | Qwen3-0.6B | Gemma3-1B | Acc | 35.53 | 31.75 | 41.35 | **45.90** |
| | | | Time | 0.29 | 0.54 | 1.04 | 0.30 |
| | Qwen3-0.6B | Qwen2.5-Math-1.5B | Acc | 35.53 | 39.86 | 43.71 | **46.13** |
| | | | Time | 0.29 | 8.71 | 6.60 | 0.27 |
| | Qwen3-0.6B | Qwen2.5-Coder-0.5B | Acc | 35.53 | 25.09 | 39.74 | **46.89** |
| | | | Time | 0.29 | 0.26 | 1.59 | 0.27 |
| Swap | Qwen2.5-0.5B | Qwen3-0.6B | Acc | 38.42 | 35.53 | 32.12 | **43.47** |
| | | | Time | 0.34 | 0.29 | 0.98 | 0.21 |
| | Qwen3-0.6B | Qwen2.5-0.5B | Acc | 35.53 | 38.42 | 41.03 | **46.50** |
| | | | Time | 0.29 | 0.34 | 1.52 | 0.26 |

Table 7: Comparison of Receiver-only, Sharer-only, T2T, and C2C across accuracy and time. The pairs are grouped into *heterogeneous* settings (where the Receiver is paired with Sharers of different capabilities) and *swap* settings (where Receiver and Sharer roles are exchanged).

## 4.4 ABLATION STUDY

**Sources of improvement**. In Table 6, we ablate the source of C2C performance gain by fixing the Receiver (Qwen3-0.6B) and varying the Sharer. *Single* denotes standard full fine-tuning of the Receiver without Sharer. *Identical* denotes C2C where both Sharer and Receiver are Qwen3-0.6B. Our default C2C uses Qwen2.5-0.5B as the Sharer. Under the same training configuration, C2C consistently attains higher accuracy than both *Single* and *Identical*. This confirms that C2C improvements do not purely come from added trainable capacity or overfitting to the training set. Instead, it points to complementary contextual understanding contributed by the heterogeneous Sharer. *Identical* still outperforms *Single*, indicating that cache-level self-communication can provide useful auxiliary understanding, echoing effects observed in latent reasoning and looped transformers (Saunshi et al., 2025; Fu et al., 2025b).

**Fuser architecture**. In Table 8 we show the effect of different components in the C2C design. Compared with pure projection that discards the Receiver's cache, fusing both models' KV-Caches and retaining the Receiver's via residual connection increases accuracy by 24.18%. Adding a gate for fused layer selection further increases the average accuracy by 3.07%.

## 4.5 BEHAVIOR ANALYSIS

**Effective rank analysis**. We analyze the effective rank of KV-Cache before and after cache-to-cache communication. Effective rank (Roy & Vetterli, 2007) is a common measure of the intrinsic dimen-

| Method | MMLU | ARC-C | OpenBook | CEval | Average |
|---|---|---|---|---|---|
| Project | 20.01 | 19.57 | 21.80 | 21.41 | 20.70 |
| +Fuse | **43.36** | 51.65 | 47.60 | 36.91 | 44.88 |
| +Gate (=C2C) | 42.92 | **54.52** | **52.60** | **41.77** | **47.95** |

Table 8: Ablation of C2C fuser components. *Project*: directly replacing the Receiver's KV-Cache with projected Sharer cache. +*Fuse*: fusing both caches and adding the result back to Receiver's via residual. +*Gate*: adding per-layer learnable gating.

sionality of model weights or activation values; a higher intrinsic dimension means richer semantic information, as formalized in Appendix A.4.1. As Table 2 shows, after cache-to-cache fusing, the effective ranks of K and V increased from 388 to 395 and from 532 to 560, respectively. This indicates that C2C enriches the semantic space by successfully transforming the Sharer's representations and injecting knowledge into the Receiver model.

**Accuracy breakdown**. Our analysis reveals that the source of C2C's accuracy gains depends on the relative capacity of the communicating models and varies across task subcategories. Details can be found in Appendix A.2.3. We also find that in some specific cases, C2C may fail as the contextual understanding from the Sharer model is not always accurate and can mislead the Receiver into generating the wrong answer. Representative example is provided in Appendix A.4.6.

**Progressive behavior**. We analyze the progressive behavior of C2C by gradually increasing the percentage of context KV-Cache being updated by C2C (details in Appendix A.2.4). When the percentage is above 50%, increasing the percentage continuously yields better performance.

**Gate behavior**. We analyze the behavior of C2C 's learnable gates under different training regimes in Appendix A.4.2. We can draw the conclusion that general-purpose training favors broad gate activation with fine-grained modulation via weights, whereas task-specific training favors sparse gate activation with stronger reliance on the selected layers.

## 5 DISCUSSION

**Future work**. C2C opens several directions for future research. (1) Cache communication in multi-agent systems: C2C may serve as a better communication primitive for real-world agentic tasks with complex multi-round reasoning, coding, and tool use. A preliminary case study on mathematical problem solving is provided in Appendix A.5.3. (2) Cross-modal collaboration: beyond text-only models, fusing caches among vision–language models (VLMs) and vision–language–action (VLA) models may enable richer multi-modal collaboration. (3) Inference acceleration: C2C can enhance speculative decoding and enable token-level routing across heterogeneous models for lower latency and cost. (4) Privacy-aware collaboration: LLMs can transmit KV-Cache segments without explicit text, limiting content exposure and improving privacy.

**Limitations**. (1) Multi-LLM systems experience performance degradation when a much weaker Sharer provides noisy information to a stronger Receiver. This limitation is shared by both T2T and C2C, as Sharer semantic quality directly impacts Receiver performance. (2) While pairwise KV-Cache communication is feasible and advantageous, scaling up the number of communicating LLMs with $O(N)$ training cost remains an open problem. Preliminary efforts towards this goal are in Appendix A.5.1 and A.5.2.

## 6 CONCLUSION

We demonstrate that LLMs can communicate beyond text. We introduce Cache-to-Cache (C2C), a general paradigm that transforms and fuses key–value (KV) caches across models to enable direct semantic communication. Across diverse tasks and model configurations, C2C consistently achieves higher task performance and better efficiency than text-to-text communication. These results establish cache-to-cache as a practical alternative to token-based communication and highlight its promise for scalable, low-latency multi-LLM systems.

## ACKNOWLEDGMENTS

This work was supported by National Natural Science Foundation of China (No. 62506197, 62325405, 62104128, U19B2019, U21B2031, 61832007, 62204164, 92364201), Tsinghua EE Xilinx AI Research Fund, and Beijing National Research Center for Information Science and Technology (BNRist). We thank Xuefei Ning for her valuable discussions and suggestions.

## ETHICS STATEMENT

This study raises no ethical issues. No human subjects or sensitive personal data were involved in the experiments.

## REPRODUCIBILITY STATEMENT

We provide sufficient information to allow the results reported in this paper to be reproduced. All experiments were conducted using publicly available datasets along with open-source models and code. Implementation details, including data selection, model architectures, hyperparameters, and training procedures, are provided in Appendix A. The code, configuration files, and model checkpoints are released at `https://github.com/thu-nics/C2C`.

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

# A APPENDIX

## A.1 DESIGN CHOICE EXPLORATION

We detail the design of C2C and discuss alternative design choices.

### A.1.1 LAYER ALIGNMENT

**Terminal alignment**. In this strategy, the layers of the two models are aligned starting from the output side. Specifically, the final layer of the smaller model is paired with the final layer of the larger model, the penultimate layer with the penultimate layer, and so on. This scheme prioritizes alignment between the deeper layers across models, which typically capture higher-level semantic representations.

**Depth-normalized alignment**. In this strategy, both models' layer indices are normalized to $[0, 1]$ by dividing by $(L-1)$, where $L$ is the total number of layers in the model. Let the model with fewer layers ($L_{\min}$) serve as the anchor. For each anchor layer $i$ (with normalized index $i/(L_{\min}-1)$), we select the layer $j$ in the other model ($L_{\max}$) whose normalized index $j/(L_{\max}-1)$ is closest:

$$j^{\star} = \arg\min_{j}\left|\frac{i}{L_{\min}-1} - \frac{j}{L_{\max}-1}\right|. \tag{5}$$

This method produces an alignment that distributes correspondences approximately uniformly across the model depth.

**C2C Choice**. In our design, we adopt **terminal alignment**, as it provides a simpler and more direct layer mapping strategy that empirically performs slightly better in our experiments.

### A.1.2 TOKENIZATION ALIGNMENT

For dialogue inputs, we first apply the chat template of each tokenizer, which produces a sequence consisting of alternating sections of (1) *template tokens* and (2) *message tokens*. These two types of sections are handled differently during alignment.

**Template sections**. Template tokens are structural markers (e.g., role delimiters, formatting tokens) that differ across tokenizers and carry no semantic content. To preserve sequence consistency without introducing unnecessary distortions, these sections are aligned by simple length padding: the shorter side is padded with <pad> tokens until both tokenizers' sequences are of equal length.

**Message sections**. Message tokens correspond to the actual textual content of user or assistant dialogs. Each target model token in a message section is decoded into its string form and re-encoded using the source model tokenizer. Special tokens (e.g., <pad>, <eos>) are mapped directly if possible; otherwise, the source model's unknown token is used. For regular tokens, if the re-encoding produces a single source model token, a direct one-to-one mapping is established. If multiple source model tokens are produced (a one-to-many case), one of the two selection strategies is applied: (1) *first-occurrence selection*: choose the first source model token from the candidate set, yielding a deterministic and computationally efficient mapping. (2) *Maximal-coverage selection*: decode each candidate token, compute its string length, and select the longest; this heuristic aims to preserve maximal surface correspondence with the original target model token.

**C2C choice** We observed that the two selection strategies generally produce very similar results, with more than 80% of sequences yielding identical alignments across strategies. Based on this observation, we empirically adopt **Maximal-coverage selection** as the default strategy to reduce the risk of losing information in one-to-many tokenization cases.

Through this design, template sections are aligned structurally via padding, while message sections are aligned semantically at the token level, ensuring robust correspondences between target model and source model representations in chat-formatted inputs.

### A.1.3 FUSER ARCHITECTURE

Beyond the C2C fuser, we also examined a more complex yet potentially more powerful variant, which we denote as **C2C-C (Complex)**. The main complexity comes from the introduction of an

Table 9: Comparison of the default C2C fuser and the complex C2C-C variant (Sharer: Qwen3-4B, Receiver: Qwen3-0.6B). PGR (Performance Gap Recovered) measures the fraction of the accuracy gap between the Receiver and Sharer that is recovered.

| Method | C-Eval | | | ARC-C | | | MMLU-Redux | | | OpenBook | | |
|---|---|---|---|---|---|---|---|---|---|---|---|---|
| | Acc | PGR | Time | Acc | PGR | Time | Acc | PGR | Time | Acc | PGR | Time |
| Qwen3-4B | 68.09 | 100% | 0.24 | 87.48 | 100% | 0.24 | 71.38 | 100% | 0.24 | 79.40 | 100% | 0.25 |
| Qwen3-0.6B | 32.04 | 0% | 0.18 | 41.04 | 0% | 0.19 | 35.53 | 0% | 0.18 | 39.20 | 0% | 0.21 |
| T2T | 36.96 | 14% | 0.92 | 52.00 | 24% | 0.80 | 42.95 | 21% | 0.99 | 46.40 | 18% | 1.70 |
| C2C | 44.40 | 34% | 0.27 | 60.17 | 41% | 0.27 | 45.92 | 29% | 0.27 | 55.20 | 40% | 0.28 |
| C2C-C | 60.63 | 79% | 0.21 | 80.96 | 86% | 0.23 | 62.78 | 76% | 0.15 | 70.40 | 78% | 0.26 |

additional projection stage: instead of directly concatenating Sharer and Receiver caches as in C2C, Sharer cache is first projected into the receiver's dimensionality through a 3-layer MLP. The concatenated representation is then processed along two familiar routes—feature fusion and dynamic weighting—to yield the final S&R cache.

The main experiment results are presented in Table 9. Note that we fix the maximum response length to 8 tokens and the maximum communication length to 256 tokens in this experiment to reduce evaluation cost. C2C-C attains stronger performance than the default C2C, suggesting that increasing the architectural sophistication of fuser can further amplify the benefits of C2C communication. In this table, we also report Performance Gap Recovered (PGR) (Ong et al., 2024) metric, which quantifies how much of the performance gap between a weak and a strong model is recovered. Nevertheless, the focus of this work is on introducing the C2C paradigm itself. For this purpose, we adopt a simple yet effective fuser design, leaving systematic investigation of more elaborate architectures to future work.

## A.2 ADDITIONAL EXPERIMENTAL RESULTS

### A.2.1 CACHE ENRICHMENT DETAIL

In Table 10 we show the effect of single-layer cache enrichment. Layer 4 and 16 benefit from the cache enrichment approach by replacing the KV-Cache with the few-shot one, while cache enrichment on other layers shows performance degradation.

Table 10: Accuracy (%) when enriching only a single transformer layer's KV-Cache via the cache enrichment oracle. Baseline without enrichment: 58.42%. Bold indicates layers that benefit from enrichment.

| Layer | Acc. | Layer | Acc. | Layer | Acc. | Layer | Acc. |
|---|---|---|---|---|---|---|---|
| 0 | 56.36 | 7 | 56.82 | 14 | 54.24 | 21 | 57.74 |
| 1 | 56.36 | 8 | 55.01 | 15 | 58.06 | 22 | 57.23 |
| 2 | 57.14 | 9 | 56.78 | 16 | **58.45** | 23 | 55.22 |
| 3 | 57.53 | 10 | 55.29 | 17 | 57.88 | 24 | 55.75 |
| 4 | **58.52** | 11 | 57.05 | 18 | 57.21 | 25 | 56.16 |
| 5 | 56.45 | 12 | 55.04 | 19 | 56.71 | 26 | 55.79 |
| 6 | 54.56 | 13 | 54.83 | 20 | 55.93 | 27 | 55.01 |

### A.2.2 STRONG-TO-WEAK COMMUNICATION

Table 11 reports the results on LongBenchV1 when pairing the weak receiver Qwen3-0.6B with a much stronger sharer, Qwen3-4B, under different input lengths. Across all length regimes, C2C consistently outperforms both the receiver alone and the T2T baseline. On average, C2C achieves a 51.01% PGR over the weak-to-strong gap. These results demonstrate that in strong-to-weak settings,

Table 11: LongBenchV1 scores comparing Receiver-only, Sharer-only, T2T, and C2C in the strong-to-weak setting (Sharer: Qwen3-4B, Receiver: Qwen3-0.6B) across different input lengths.

| Length | Receiver | Sharer | T2T | C2C |
|--------|----------|--------|-------|-------|
| 0–4k | 30.52 | 50.48 | 38.52 | 41.46 |
| 4–8k | 26.03 | 48.28 | 35.79 | 37.57 |
| 8k+ | 25.99 | 44.36 | 33.79 | 34.23 |
| Average | 27.63 | 47.90 | 36.18 | 37.97 |

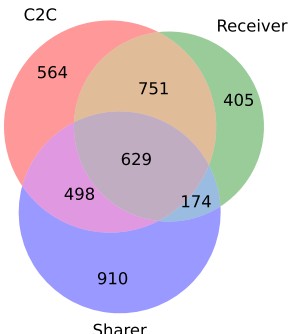
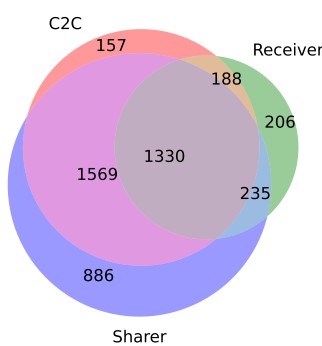

(a) Sharer: Qwen2.5-Math-1.5B, Receiver: Qwen3-0.6B   (b) Sharer: Qwen3-4B, Receiver: Qwen3-0.6B

Figure 7: Venn diagrams of correctly answered questions on MMLU-Redux, comparing Receiver-only, Sharer-only, and C2C (C2C-C variant) answer sets.

C2C can effectively transfer the stronger model's contextual understanding, yielding notable gains for the weaker receiver.

We additionally evaluated the strong-to-weak setting (Qwen3-0.6B as receiver and Qwen3-4B as sharer) on other benchmarks beyond LongBenchV1. The detailed results are provided in Section A.1.3, Table 9.

### A.2.3 ACCURACY BREAKDOWN

We analyze where the accuracy gains of C2C come from by using Venn diagrams on the MMLU-Redux benchmark, as illustrated in Figure 7. For this analysis, we use the C2C-C variant introduced in Section A.1.3, as it has the potential to achieve stronger performance and provides a clearer breakdown of where C2C 's accuracy originates.

**Models with comparable capacity**. When the Receiver (Qwen3-0.6B) and the Sharer (Qwen2.5-Math-1.5B-Instruct, denoted as Qwen2.5-Math-1.5B) have comparable overall capacity but complementary strengths, C2C not only inherits part of the Sharer's ability but also solves additional questions by integrating understanding from both models.

**Models with disparate capacity**. When the Sharer (Qwen3-4B) is substantially stronger than the Receiver (Qwen3-0.6B), C2C tends to integrate more of the stronger model's understanding. Quantitatively, in the disparate-capacity case (Figure 7b), among the questions that the Sharer can answer correctly, C2C also answers 72.11% correctly. In contrast, in the comparable-capacity case (Figure 7a), C2C succeeds on only 50.97%.

**Subcategory breakdown**. We analyze the accuracy gains on different subcategories using the radar plot and histogram plot shown in Figure 8, Figure 9, and Figure 10. The result shows that for the Qwen2.5-0.5B and Qwen3-0.6B model pair, C2C outperforms T2T on 12 out of 17 total categories. Using C2C in history, law, and chemistry can bring an additional 7, 7.6, and 7.5 accuracy improvement compared to T2T. For the Qwen3-4B and Qwen3-0.6B model pair, C2C outperforms T2T on all 17 categories. Using T2T in engineering results in a 1% performance drop, showing the case where text communication between models fails to help the receiver models achieve better

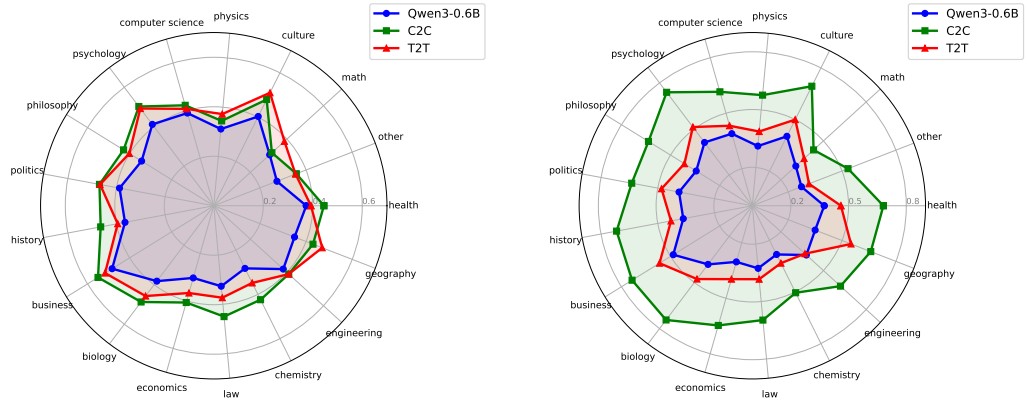

(a) Sharer: Qwen2.5-0.5B, Receiver: Qwen3-0.6B    (b) Sharer: Qwen3-4B, Receiver: Qwen3-0.6B

Figure 8: Per-subcategory accuracy on MMLU-Redux, comparing Receiver, Sharer, T2T, and C2C.

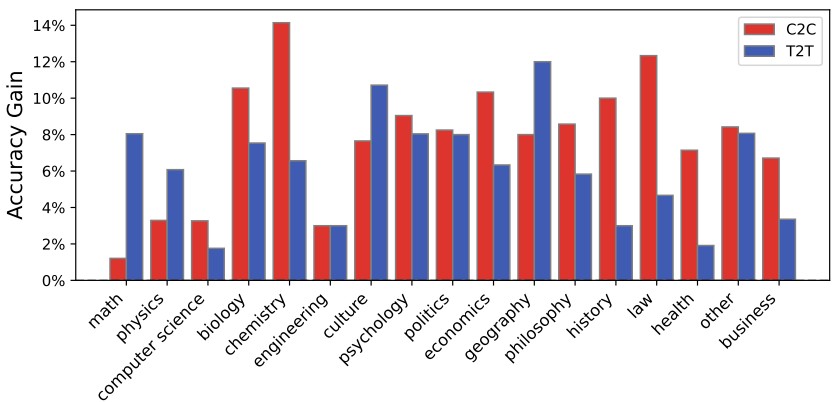

Figure 9: Per-subcategory accuracy gain of T2T and C2C over the Receiver baseline on
MMLU-Redux.
(Sharer: Qwen2.5-0.5B, Receiver: Qwen3-0.6B)

performance. At the same time, C2C better utilizes the 4B model's contextual understanding of the
problems and achieves a 24% increase in accuracy.

### A.2.4   PROGRESSIVE BEHAVIOR

To investigate the impact of fused KV-Cache proportion on the accuracy of the receiver model, we
gradually added the proportion of fused KV-Cache derived from the sharer to the receiver model be-
fore generating outputs. Specifically, former and latter refer to progressively replacing the receiver's
KV-Cache with the fused KV-Cache from front to back and back to front, respectively. We observe
that the overall accuracy first decreases and then increases as the replacement ratio grows. The per-
formance reduction may stem from the gap between training and testing, where only the full receiver
KV-Cache is used during training. When the fused proportion goes up to over 50%, the performance
of C2C continues to increase with respect to the proportion, reflecting the progressive benefits of
C2C. Note that projecting using the latter cache generally has a larger impact than projecting the
former, since it is closer to the final response.

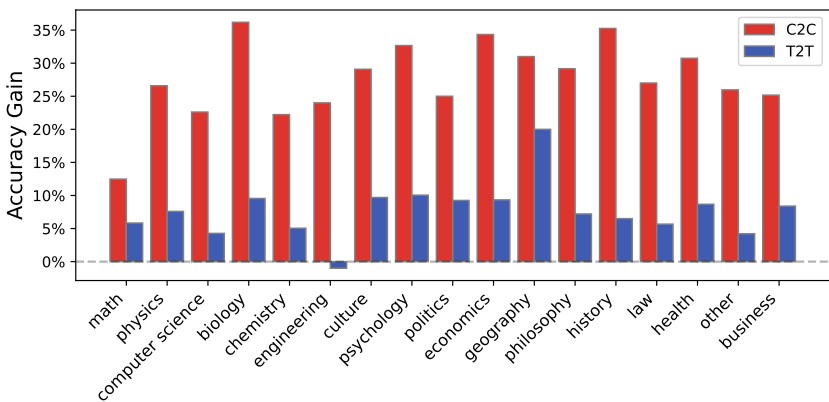

Figure 10: Per-subcategory accuracy gain of T2T and C2C over the Receiver baseline on
MMLU-Redux.
(Sharer: Qwen3-4B, Receiver: Qwen3-0.6B)

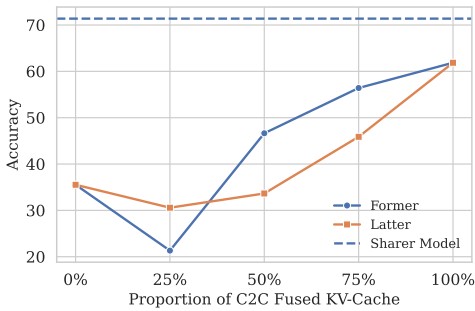

Figure 11: MMLU-Redux accuracy as the proportion of the Receiver's KV-Cache replaced by
C2C-fused cache increases. "Former" replaces tokens front-to-back; "Latter" replaces
back-to-front.
(Sharer: Qwen3-4B, Receiver: Qwen3-0.6B)

## A.3 ADDITIONAL EXPERIMENT SETUP

### A.3.1 CACHE ENRICHMENT

We conducted Oracle experiments with Qwen3-0.6B and Qwen3-4B to examine how KV-Cache
enrichment influences model performance. The evaluation was performed on MMLU-Redux. The
few-shot examples are selected from MMLU while excluding overlaps with MMLU-Redux to ensure
fairness. To probe different ways of applying cache enrichment, we compared four cache enrichment
strategies: All-layer Cache Enrichment (apply cache enrichment on all layers), Single-Layer Cache
Enrichment (apply cache enrichment only on single layers), Selective Cache Enrichment - Best (se-
lect n layers that have the highest accuracy according to Single-Layer Cache Enrichment), Selective
Cache Enrichment - Worst (select n layers that have the lowest accuracy according to Single-Layer
Cache Enrichment). All methods utilized Few-Shot–optimized KV-Caches while maintaining the
same cache length as Zero-Shot, enabling a controlled evaluation of cache enrichment and its layer-
specific effects.

### A.3.2 CACHE TRANSFORMATION

We employed the MMLU-Redux dataset to train a 3-layer MLP that maps the last layer KV-Cache
in the source LLM (Qwen3-4B) to the last layer of the target LLM (Qwen3-0.6B). In this oracle
experiment, we adopt the MSE loss of the projected KV-Cache and the target KV-Cache instead of
the next-token prediction loss, as we aim to test whether the representation space can be transformed

using a neural network. The Key Cache and Value Cache are first concatenated on the hidden space dimension, then put for MSE calculation. A detailed calculation of loss is shown in the following code:

```
source_prefilled = source_model.forward(prefill_input_id)
target_prefilled = target_model.forward(prefill_intput_id)
source_k, source_v = source_prefilled.past_key_values[-1]
target_k, target_v = target_prefilled.past_key_values[-1]
project_k = k_projector(source_k)
project_v = v_projector(source_v)
mseloss(torch.dstack([project_k, project_v]),
        torch.dstack([target_k, target_v]))
```

For visualization, 300 samples were randomly selected from the dataset. The source, target, and transformed KV-Cache were all projected into two-dimensional space using t-SNE, allowing us to examine the alignment of representations between the two models. For t-SNE generation, we set perplexity to 50 and max iterations to 1000.

### A.3.3  QUERY-LEVEL ROUTING

Query-level routing aims to improve the performance–efficiency trade-off by dynamically assigning harder queries to a stronger LLM. Following prior work, we adopt a matrix factorization framework. Query embeddings are obtained from the OpenAI text-embedding-3-small encoder, while model embeddings are taken from pretrained vectors of gpt-4-1106-preview and mixtral-8x7b-instruct-v0.1. These embeddings are used to compute a strong win rate score for each query, which reflects its relative difficulty. Queries are then ranked by this score. For each evaluated model pair, we define the strong model as the one achieving higher standalone benchmark accuracy and the weak model as the lower-performing one. Queries in the upper half of the ranking are routed to the strong model, while those in the lower half are routed to the weak model.

### A.3.4  EVALUATION METHOD

**Main evaluation**. We evaluate C2C on four multiple-choice benchmarks: OpenBookQA, MMLU-Redux, ARC-Challenge, and C-Eval. For MMLU-Redux, we exclude questions annotated with the error type *no correct answer*. For all evaluations, we adopt a deterministic generation configuration without sampling, using greedy decoding to ensure reproducibility. Specifically, we use Non-CoT prompts, following the unified format described in Section A.3.6. Model outputs are then matched to the correct option labels to compute accuracy. To control evaluation cost, we set the maximum response length to 64 tokens unless otherwise specified, where the response refers to the final answer generated by the Receiver, since the base models do not always follow instructions, and longer limits would substantially increase inference time. For the T2T setting, we additionally set the maximum communication length to 256 tokens, where the communication refers to the messages passed from the Sharer to the Receiver.

**LongBench evaluation**. We evaluate C2C on the LongBench-E dataset, which comprises a total of 13 individual datasets. The prompts and evaluation procedures use the official LongBench settings, with a maximum output length of 2,048 tokens.

### A.3.5  C2C TRAINING

**Training data**. (1) *Performance experiment.* The fuser was trained on the OpenHermes-2.5 Dataset with a maximum sequence length of 2,048 tokens. Training used 500,000 samples for one epoch with a macro batch size of 256, corresponding to 1,929 total training steps.

(2) *Scaling sequence lengths experiment.* The fuser was trained on the LongBench-E benchmark with a maximum sequence length of 12,000 tokens. The data was randomly split into 3/4 for training and 1/4 for evaluation by data index, ensuring independence between training and evaluation. Training used 1,896 samples for one epoch with a macro batch size of 16, corresponding to 118 total training steps.

(3) *Scaling model sizes and different model combinations experiment.* The fuser was trained on the auxiliary_train split of the MMLU dataset with a maximum sequence length of 1,024 tokens. Training used 15,000 samples for one epoch with a macro batch size of 128, corresponding to 116 total training steps.

**Training scheme**. All experiments were conducted with a fixed random seed of 42 to ensure reproducibility. Unless otherwise noted, the training configuration was as follows: optimization employed a learning rate of $1 \times 10^{-4}$ with a linear scheduler and a 10% warmup ratio, a weight decay of 0.01, and a maximum gradient norm of 1. The temperature was linearly annealed from 1.0 to 0.001 across the total number of training steps. Layer alignment was configured with the last aligned scheme across all experiments. The tokenization alignment was applied only when the paired models employed different tokenizers, in which case the longest strategy was used. For data preparation, each dataset was partitioned into a training split (99%) and a small held-out validation split (1%). The validation split was not used for model updates but was monitored during training to report evaluation loss.

### A.3.6 EVALUATION PROMPTS

Text 1 presents the prompts used for the main evaluation on multiple-choice datasets. Text 2 is the alternative non-CoT version used for response diagnosis in Appendix A.4.4. Text 3 provides the prompt for the Sharer model in the T2T evaluation; the Receiver model uses the same prompt as in the C2C setting. Text 4 shows the prompt used in the cache enrichment experiment. For the zero-shot method, no shots are included in the prompt. The few-shot method uses exactly the same prompt as Text 4. For Oracle methods, we adopt the few-shot prompt but remove the KV-Cache associated with the shots after the forward pass. The prompt for LongBench evaluation follows its official configuration, which varies across the different sub-datasets.

---

**Text 1. Prompt for Non-CoT Evaluation**

Accurately answer the following question:

{QUESTION}

**Choices:**
{CHOICES}

**Instructions:**
- Carefully read the question and all options.
- Select the single most correct answer.
- Respond ONLY in the following format: "The correct answer is A/B/C/D".
- Do not include any explanations, additional text, or punctuation besides the answer.
The correct answer is

---

**Text 2. Prompt for CoT Evaluation**

Accurately answer the following question:

{QUESTION}

**Choices:**
{CHOICES}

**Instructions:**
- Carefully read the question and all options.
- Let's think step by step and explain your reasoning briefly.
- Then give the final answer starting with The correct answer is.

---

> **Text 3. Prompt for Sharer model in T2T Evaluation**
>
> In one clear sentence, describe the most essential background knowledge needed to answer the question: {QUSETION} Do NOT directly solve or give answer to the question.

> **Text 4. Prompt for Oracle Experiment**
>
> The following are single choice questions (with answers) about {SUBJECT}.
>
> *Shot 1:*
>
> **Question:** {QUSETION}
>
> **Options:**
> {OPTIONS}
>
> **Answer:** {ANSWER} *( A,B,C or D)*
>
> . . .
>
> *Shot N:*
>
> **Question:** {QUSETION}
>
> **Options:**
> {OPTIONS}
>
> **Answer:** {ANSWER} *( A,B,C or D)*
>
> **Question:** {QUSETION}
>
> **Options:**
> {OPTIONS}
>
> **Answer:**

## A.4   ADDITIONAL ANALYSIS

### A.4.1   EFFECTIVE RANK

We list the definition of effective rank that was proposed by Roy & Vetterli (2007) here as a reference. For a matrix W that has size $M \times N$, the singular value decomposition of it can be expressed as $W = U\Sigma V$ and the singular values $\sigma = (\sigma_1, \sigma_2, ..., \sigma_{min(M,N)})^T$ are the non-negative diagonal entries of the matrix $\Sigma$. The singular value distribution is denoted as:

$$p_i = \frac{\sigma_i}{\|\sigma\|_1} \tag{6}$$

Denote the Shannon Entropy as:

$$H(p_1, p_2, ..., p_{min(M,N)}) = - \sum_{i=1}^{min(M,N)} p_i \log p_i \tag{7}$$

The effective rank is defined as:

$$erank(W) = e^{-\sum_{i=1}^{min(M,N)} p_i log p_i} \tag{8}$$

In Figure 12, we present the effective rank of key and value caches across all the layers. The plot shows a continuous increase in the effective rank of value caches after applying C2C, especially

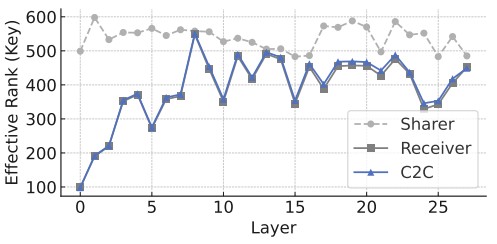

(a) Effective rank of the Key cache.

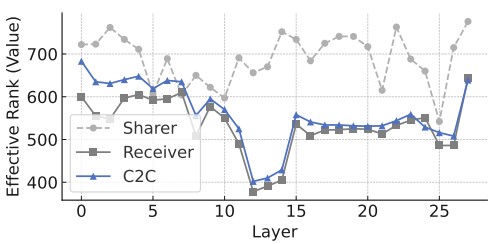

(b) Effective rank of the Value cache.

Figure 12: Per-layer effective rank of Key and Value caches for the Receiver (Qwen3-0.6B), Sharer (Qwen3-4B), and after C2C fusion. Higher effective rank indicates richer semantic information.

in the shallow layers. Key caches after applying C2C also have a comparable effective rank and increase at deep layers.

### A.4.2 GATING BEHAVIOR

We analyze the behavior of the learnable gates by contrasting models trained on general-purpose versus task-specific data. This comparison reveals markedly different gating dynamics across the two regimes.

**General-purpose training**. When C2C is trained on the OpenHermes-2.5 dataset, the learned key and value gates remain almost fully open. Across the three model combinations reported in Table 4, the average gate activation ratio exceeds 98.21%. Despite this near-complete activation, we observe that in certain layers the dynamic weights are concentrated at very small values—for example, the average key weight in some layers falls below 0.1. This suggests that under general-purpose training, C2C leverages the dynamic weighting mechanism to modulate how much information is incorporated from the sharer on a per-query basis, effectively treating dynamic weights as the primary control signal while leaving most gates open.

**Task-specific training**. In contrast, when C2C is trained on the MMLU auxiliary_train split, the gates exhibit a much sparser activation pattern. Across model combinations shown in Table 7, the average gate activation ratio drops to 52.67%. For the layers where gates do open, however, the dynamic weights are substantially larger, with most layers exhibiting average weights above 0.4. This indicates that under task-specific training, the gating mechanism selects a smaller subset of layers that are consistently useful, while the dynamic weights primarily regulate the contribution strength of these selected layers.

Overall, these findings highlight the adaptive interplay between gates and weights: general-purpose training favors broad gate activation with fine-grained modulation via weights, whereas task-specific training favors sparse gate activation with stronger reliance on the selected layers.

### A.4.3 OUTLIER CASES IN INFERENCE TIME

**Llama3.2**. We observe that the Llama3.2 model achieves significantly lower inference time compared to other baselines in Table 4. This improvement can be attributed to two factors. First, the Llama3.2 model itself has faster inference speed due to its implementation. Second, under the Non-CoT evaluation prompts described in Section A.3.6, the model tends to output only a single option letter (e.g., "A" or "B"), rather than a longer formatted string such as "The correct answer is A." The shorter outputs further reduce the average decoding time, leading to the observed advantage.

**Qwen2.5-Math**. In contrast, the Qwen2.5-Math model exhibits considerably longer inference time, as shown in Table 7. The primary cause is its tendency to ignore the Non-CoT evaluation and T2T prompts described in Section A.3.6, producing verbose, step-by-step solutions rather than concise answers. To accommodate these long outputs and avoid truncation, we set both the maximum response length and the maximum communication length to 1024 tokens during evaluation. Under this configuration, the model decodes substantially more tokens on average, resulting in significantly longer inference time.

### A.4.4 EXAMPLE MODEL OUTPUT

To help better understand the response pattern of different models, we show the example response from different methods using the CoT prompt. We use Qwen2.5-0.5B as the Sharer model and Qwen3-0.6B as the Receiver model.

As shown in the following examples, Qwen3-0.6B misunderstood the question. It treated a probe for deeper meaning as merely a rephrasing task. Qwen2.5-0.5B-Instruct misinterpreted the question type altogether, taking it as sentence completion and answering the replaced question instead, with overly verbose reasoning. T2T altered the context of the question itself. In contrast, C2C maintained a solid understanding, recognizing that the rewritten sentence aimed to explore deeper moral reasoning, and accordingly selected the correct answer.

---

**Text 5. Question**

Instead of asking why the act of destroying the environment might be immoral, Hill wants to ask . . .

**Choices:**

    A) why the act of destroying nature might be immoral.

    B) why people who destroy the environment might be bad people.

    C) how the decision to preserve the environment benefits the environment.

    D) whether plants have interests.

---

**Text 6. Qwen3-0.6B Response**

The correct answer is A. why the act of destroying nature might be immoral.
Hill is not asking why the act of destroying the environment is immoral, but rather what he wants to ask in a different way. The other options do not align with the question as stated.
The correct answer is A.

---

**Text 7. Qwen2.5-0.5B-Instruct Response**

To accurately answer the question, let's carefully analyze each option and think through the implications of each choice:
A. This option suggests that the act of destroying nature might be considered immoral. This is a reasonable and common interpretation of the act of destroying nature, as it involves the destruction of natural resources and ecosystems. However, it does not directly address the question of why the act might be immoral.
B. This option implies that people who destroy the environment are inherently bad. This is a more extreme interpretation and does not align with the general understanding of environmental destruction. It also does not directly address the question of why the act might be immoral.
C,D. ...
After analyzing all the options, the most appropriate and accurate answer is:
The correct answer is A. Why the act of destroying nature might be immoral.
This answer directly addresses the question by suggesting that the act of destroying nature might be considered immoral, which aligns with the general understanding of environmental destruction.

**Text 8. Text-to-Text Response**

**Context from Qwen2.5-0.5B:.**

To understand the concept of environmental destruction, one must first grasp the fundamental principles of sustainability, the importance of biodiversity, and the impact of human activities on the natural world.

**Response from Qwen3-0.6B:.**

The correct answer is **C. how the decision to preserve the environment benefits the environment**.
Hill wants to ask how the act of preserving the environment benefits the environment, not why it might be immoral. This aligns with the question's focus on understanding the consequences of environmental action rather than its moral implications.
The other options either directly address the question or are irrelevant.
The correct answer is C.

**Text 9. C2C Response**

The correct answer is B. Hill wants to ask why people who destroy the environment might be bad people. Hill's goal is to understand the motivations behind people's actions and to identify the underlying reasons for their destructive behavior. This can help in developing strategies to prevent such actions and promote sustainable practices. Hill's question is a way to explore the complexities of human behavior and to gain a deeper understanding of the factors that contribute to destructive actions. The other options are not the best answers because they are not directly related to the question or the reasons behind destructive behavior. The correct answer is B.

### A.4.5 TRAINING COST ANALYSIS

Here we present the training cost in Table 12. As shown in the table, accuracy quickly increases in the first few hundred steps. With 300 training steps (less than 9 GPU hours), C2C achieves a comparable result to the final checkpoint.

Additionally, in Figure 13 we have plots of training loss and validation loss to compare the settings using Qwen2.5-0.5B or Llama3.2-1B as Sharer. The plot shows that for both settings, the training loss converges at 250 steps and the eval loss becomes stable at step 1000, suggesting that C2C across different model families incurs no extra computational cost.

### A.4.6 FAILURE MODE ANALYSIS

We have shown that in various settings, C2C, as a new communication paradigm, can increase the receiver model performance by using the sharer's contextual understanding. However, in some specific cases, C2C can fail to do so as the contextual understanding from the Sharer model is

Table 12: Training cost (wall-clock and GPU hours) and MMLU-Redux accuracy at each training step (columns) for three Sharer–Receiver pairs.

| C2C Setting | Metric | 0 | 50 | 100 | 150 | 300 | 1929 |
|---|---|---|---|---|---|---|---|
| **Qwen2.5-0.5B+Qwen3-0.6B** | Wall-clock (h) | - | 0.15 | 0.29 | 0.44 | 0.87 | 5.59 |
| | GPU Hours | - | 1.20 | 2.32 | 3.52 | 6.95 | 44.72 |
| | MMLU-Redux | 35.53 | 40.36 | 41.10 | 42.06 | **44.30** | 42.92 |
| **Llama3.2-1B+Qwen3-0.6B** | Wall-clock (h) | - | 0.18 | 0.35 | 0.53 | 1.05 | 6.78 |
| | GPU Hours | - | 1.44 | 2.80 | 4.24 | 8.44 | 54.24 |
| | MMLU-Redux | 35.53 | 28.66 | 41.16 | 43.98 | 44.37 | **44.42** |
| **Qwen3-4B-Base+Qwen3-0.6B** | Wall-clock (h) | - | 0.16 | 0.32 | 0.48 | 0.96 | 6.17 |
| | GPU Hours | - | 1.28 | 2.56 | 3.84 | 7.68 | 49.36 |
| | MMLU-Redux | 35.53 | 38.07 | 39.56 | 40.77 | **44.11** | 43.95 |

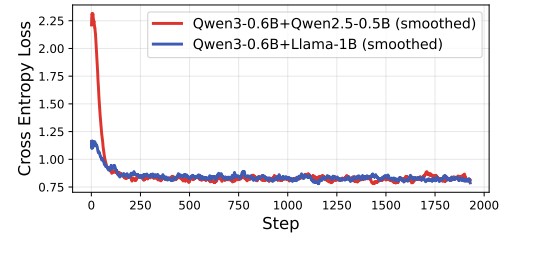 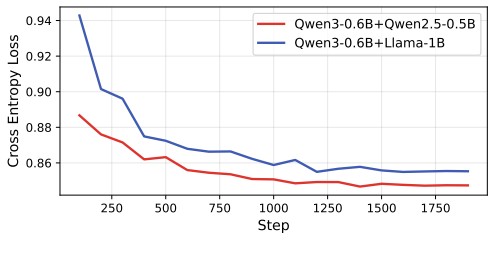

(a)                     Training loss comparison                (b) Validation loss comparison

Figure 13: Training and validation loss curves for C2C with two Sharer–Receiver pairs: Qwen2.5-0.5B+Qwen3-0.6B and Llama3.2-1B+Qwen3-0.6B.

Table 13: MMLU accuracy in the one-Receiver, multiple-Sharers setting (Receiver: Qwen3-0.6B, Sharer 1: Qwen3-4B-Base, Sharer 2: Qwen2.5-1.5B-Math). Arrows denote KV-Cache transfer direction via C2C.

| Type | MMLU Acc. |
|---|---|
| Receiver | 35.53 |
| Sharer1 | 1.03 |
| Sharer2 | 39.86 |
| Sharer1 → Receiver | 60.71 |
| Sharer2 → Receiver | 46.13 |
| Sharer1, Sharer2 → Receiver | 64.60 |

not always accurate and can mislead the Receiver into generating wrong answers. As shown in Figure 7, though in general the number of correctly answered questions increases, there are some questions that the receiver can answer correctly but are answered incorrectly after applying C2C. Below we show an example of an accounting question where Qwen3-0.6B generates the correct answer. However, when using T2T and C2C, Qwen2.5-0.5B conveys incorrect information, leading Qwen3-0.6B to the wrong answer.

This type of failure is more likely when using a very weak Sharer with strong Receiver and can lead to a performance degradation as mentioned in the limitation part.

---

**Text 10. Question**

Sussman Co. prepared cash-basis financial statements for the month ended January 31. A summary of Sussman's January activities follows: Credit sales of $5600. Collections of $1900 relating to January credit sales. Accrued salaries of $1200. By what amount will Sussman's cash-basis income for the month ended January 31 increase as a result of restating these activities to the accrual basis of accounting?

**Choices:**

    A) $2,500

    B) $3,700

    C) $4,400

    D) $4,900

## Text 11. Qwen3-0.6B Response

To determine the increase in cash-basis income for Sussman Co. as a result of restating its January activities to the accrual basis, we need to identify the **cash** and **accrual** differences.
- **Credit sales**: $5,600 (cash is not recorded yet) - **Collections**: $1,900 (cash is recorded) - **Accrued salaries**: $1,200 (cash is not recorded yet)
**Cash-basis income** would be: $5,600 (credit sales) + $1,900 (collections) - $1,200 (accrued salaries) = **$5,300**
**Accrual basis income** would be: $5,600 (credit sales) + $1,900 (collections) + $1,200 (accrued salaries) = **$7,700**
**Increase in cash-basis income = $7,700 - $5,300 = $2,400**
The correct answer is: **A. $2,500**.

## Text 12. Qwen2.5-0.5B-Instruct Response

To determine the increase in Sussman's cash-basis income due to restating the January activities to the accrual basis of accounting, let's follow these steps:
1. **Identify the cash-basis income for January:** - Credit sales: $5600 - Collections: $1900 - Accrued salaries: $1200 - Total cash-basis income: $5600 + $1900 + $1200 = $8700
2. **Understand the impact of restating activities to the accrual basis:** ...
3. **Calculate the understated cash-basis income:** ...
- Understated cash-basis income = $8700 - $5600 = $3100
4. **Determine the increase in cash-basis income:** - The increase in cash-basis income is the understated cash-basis income. - Increase in cash-basis income = Understated cash-basis income = $3100
5. **Match the increase to the given choices:** - The correct answer is $3100, which corresponds to choice B.
Therefore, the correct answer is B. $3,700.

## Text 13. Text-to-Text Response

**Context from Qwen2.5-0.5B:**.

To determine the increase in Sussman's cash-basis income for the month ended January 31 due to restating the activities to the accrual basis of accounting, you need to consider the following steps:
1. **Identify the cash-basis income for January 31**: ...
2. **Identify the accrual basis income for January 31**: ...
3. **Calculate the difference between the accrual basis and cash-basis income**: ...

**Response from Qwen3-0.6B:**.

The correct answer is **B. $3,700**.
To determine the increase in Sussman's cash-basis income for the month ended January 31 due to restating the activities to the accrual basis of accounting, we need to calculate the difference between the accrual basis and cash

## Text 14. C2C Response

To calculate the increase in cash-basis income for the month ended January 31, we need to subtract the cash sales from the credit sales. In this case, the cash sales are $1900 and the credit sales are $5600. So, the increase in cash-basis income is $5600 - $1900 = $3700. The correct answer is B.

Table 14: MMLU-Redux accuracy in the multiple-Receiver, multiple-Sharer setting with shared latent projectors. Receivers: Qwen3-0.6B and Qwen2.5-0.5B-Instruct; Sharers: Qwen2.5-Coder-0.5B-Instruct and Qwen3-4B-Base.

| Sharer | Receivers (MMLU Acc.) | |
| --- | --- | --- |
| | Qwen3-0.6B | Qwen2.5-0.5B |
| None | 34.78 | 38.44 |
| Coder-0.5B | 46.07 | 42.44 |
| 4B-Base | 51.23 | 59.17 |
| Both | 51.08 | 59.81 |

### A.5 C2C Extensions and Future Work

As an initial exploration, this work focuses on pairwise communication to establish a foundation for future research. Communication via KV-Cache among multiple models is a direction that requires future exploration, with different designs that can be employed. Here, we explore two settings that involve multiple Sharers and/or Receivers.

#### A.5.1 One Receiver, Multiple Sharers

We explore the scenario in which three models are involved in the communication. Here we set Qwen3-0.6B as the Receiver and use both Qwen2.5-1.5B-Math and Qwen3-4B-Base as Sharers. Here, we separately trained the C2C fusers from Qwen2.5-1.5B-Math to Qwen3-0.6B and Qwen3-4B-Base to Qwen3-0.6B. The result in Table 13 shows that without additional training, the receiver can use C2C to fuse two Sharers' semantic information and improve the performance.

#### A.5.2 Multiple Receivers Multiple Sharers

We provide a possible path towards more efficient scaling, with $O(N)$ fusers instead of $O(N^2)$ fusers to communicate among $N$ LLMs.

The design philosophy is to decompose $M \times N$ communication routes into $M \times 1$, then $1 \times N$ steps, with 1 being the unified latent space. This is achieved by using Projectors $P_{s_i}$ to transform the KV-Cache from different Sharer LLMs $s_0, \ldots, s_{M-1}$ into a unified latent representation KV-Latent $L$, then using the C2C fuser $F_{r_j}$ to fuse them with each Receiver model $r_0, \ldots, r_{N-1}$. $C_{s_i}$ represents the KV-Cache from model $s_i$. For M Sharers, N Receivers, we have M projectors and N fusers. Each sharer-receiver pair has its own gate. During training, all Sharer KV-Caches are projected into a unified hidden space with respective Projectors.

$$C_{s_i} \xrightarrow{P_{s_i}} U_{s_i}, \quad i = \mathbb{M} = \{0, \ldots, M-1\}$$

Then, each of the $N$ Receivers fuses all the $M$ available information from this hidden space.

$$(U_{s_i}, C_{r_j}) \xrightarrow{F_{r_j}} C_{r_j}, \quad j = \mathbb{N} = \{0, \ldots, N-1\}$$

This yields $N$ updated caches $C'_{r_j}$, which can be optimized using the same SFT loss discussed in Section 3.3.4. Losses from all Receivers are summed to update all Projectors and fusers. During inference, the model supports both one-to-one and many-to-one (multi-sharer to single receiver) settings. As an initial exploration, we directly adopted the training data, fuser architecture, and hyperparameter settings from the pairwise fuser training, with vast possibilities left for future work.

In Table 14 we show the result of using Qwen2.5-Coder-0.5B-Instruct and Qwen2.5-Math-1.5B-Instruct as Sharers, Qwen3-0.6B and Qwen2.5-0.5B-Instruct as Receivers.

#### A.5.3 Incorporating with Agentic Flow

Here we show an exploration of C2C in agentic flow. We adopt the agentic flow for math problem-solving, following the workflow and prompts from Cognify (He et al., 2025). The workflow consists of a problem interpreter that analyzes the math problem and a problem solver that calculates the

Table 15: GSM8K accuracy comparing single-model inference, multi-agent T2T flow, C2C, and T-C2C (T2T interpretation combined with C2C cache fusion).

| Method | Accuracy |
|---|---|
| Single Model (Qwen3-0.6B) | 41.17 |
| T2T (Multi-agent flow) | 61.18 |
| C2C | 62.55 |
| T-C2C | 78.01 |

answer. For C2C, we use the prompt of the math solver, so the problem interpretation is done by KVCache fusion. Here we present the result on GSM8K. As shown in the table, the multi-agent flow increases the accuracy of math problem solving by 20.01. C2C has an accuracy of 62.55, increasing the accuracy by 1.37 when compared with the multi-agent flow using T2T. The performance can be further elevated to 78.01 by using T-C2C, which lets the math interpreter generate the interpretation and use C2C on both the question and the interpretation.

