# OpenReview forum: "Cache-to-Cache: Direct Semantic Communication Between Large Language Models"
_ICLR.cc/2026/Conference — ICLR 2026 Poster_

### Official Review · Reviewer_qYJq · 2025-10-28

**Soundness:** 3
**Presentation:** 3
**Contribution:** 3
**Rating:** 6
**Confidence:** 3

**Summary:**

The paper “Cache-to-Cache: Direct Semantic Communication Between Large Language Models” proposes a new paradigm (C2C) that lets multiple LLMs exchange information directly via their KV-caches instead of text. By projecting and fusing one model’s cache into another through a learnable neural “Fuser” with gating, C2C enables richer, faster inter-model communication that preserves deep semantics. Experiments show 3–5 % higher accuracy and about 2× speedup over text-to-text interaction. It generalizes across model families and sizes, outperforming routing and text collaboration baselines while supporting heterogeneous model cooperation.

**Strengths:**

1. Novel idea of transferring cache with little restriction on the type of model.
2. Does not require change of the receiver or sender.
3. Shows improvement over text to text and cache to cache baselines.
4. Proposes an interesting argument. Although the reviewer personally do not support this claim (see comment below), I think the initial results are nice.

**Weaknesses:**

1. Lack of experiments on bigger models.
2. The theory is lacking. To me, there is an implicit assumption that generating KV cache from text is more costly than the fusing operation introduced. However, transforming between two different model classes is intuitively a more complex operation. Can authors explain why this might be cheaper than KV recomputation?

**Questions:**

1. What is the training overhead for the fuser? Transforming a llama 3.2 1B's KV into Qwen 0.5B's KV seems to be a computational heavy idea.
2. Is this idea reasonable? If you have two models of different architectures, transferring the KV cache does not seem like a good idea to me. The main reason is that the difference in the numeric value of KV are probably too high such that transforming one into another will be more difficult than generating the KV from text.

---

> ### Author Response · Authors · 2025-11-23
>
> We sincerely thank Reviewer qYJq for the positive feedback.
> We appreciate your valuable recognition of our **novelty, communication scheme, and performance benefits**.
> We address all the concerns and questions below.
>
> ### W1: Larger model
>
> > Lack of experiments on bigger models.
>
> We perform scaling experiments across 0.5B to 14B models on the MMLU dataset, as shown in Figure 6 of the paper.
> Since C2C consistently outperforms T2T across this wide range of sizes, we believe it demonstrates strong scalability. We agree that experiments on even larger models (e.g., 32B+) are a valuable direction for future work.
>
> ### W2: Cost comparison between KV-Cache generation and C2C
>
> > The theory is lacking. To me, there is an implicit assumption that generating KV cache from text is more costly than the fusing operation introduced. However, transforming between two different model classes is intuitively a more complex operation. Can authors explain why this might be cheaper than KV recomputation?
>
> We clarify that the cost of KV-Cache generation differs significantly between prefilling (parallel) and decoding (sequential), as evident in commercial pricing (e.g., GPT-5 input tokens are $8\times$ cheaper than output tokens [1]).
>
> The reviewer is correct that transforming KV-Cache is more complex than prefilling text.
> However, T2T communication requires decoding the text message first, which is the primary bottleneck. C2C bypasses this expensive sequential decoding entirely, replacing it with a parallel KV-Cache fusion.
> We illustrate this in the runtime breakdown below. Here, we use Qwen3-0.6B as the Receiver and Qwen2.5-0.5B as the Sharer.
>
> |Metric|Receiver|Sharer|T2T|C2C|
> |:--|:--:|:--:|:--:|:--:|
> |Input Token|170|187|103+332|170|
> |Output Token|11|19|10|12|
> |Communication Token|0|0|80|0|
> |-|-|-|-|-|
> |Prefill (ms)|27|20|21+32|47|
> |Decode (ms)|281|326|1312+231|308|
> |Fusion (ms)|0|0|0|90|
> |-|-|-|-|-|
> |Total Latency (ms)|308|346|1596|445|
>
> T2T spends 1312ms decoding the communication text. In contrast, C2C fusion takes only 90ms. This massive reduction in communication latency far outweighs the cost of the fusion operation itself, resulting in a $>3\times$ total speedup.
>
> [1] https://openai.com/api/pricing/
>
> ### Q1: C2C training cost
>
> > What is the training overhead for the fuser? Transforming a llama 3.2 1B's KV into Qwen 0.5B's KV seems to be a computational heavy idea.
>
> We wish to clarify that C2C does not *replace* the Receiver's KV-Cache. Instead, it extracts and projects useful information from the Sharer into the Receiver's space, *adding* it via a residual connection.
>
> In the Appendix, we also add training dynamics for Qwen2.5-0.5B and Llama3-1B Sharers.
> The plots show that for both settings, training loss converges before 300 steps and eval loss stabilizes at around 1000 steps, confirming that cross-family transformation does not incur extra computational overhead.
>
> To quantify this, we present the detailed training cost table. We employ pair-specific training due to its manageable cost and strong downstream generalizability. To quantify this, we present the training dynamics on 8 NVIDIA A800 GPUs in the table below.
>
> | Model Pair | Metric | 0 (Baseline) | 50 Steps | 100 Steps | 150 Steps | 300 Steps | Final (1929) |
> |:---|:---|:---:|:---:|:---:|:---:|:---:|:---:|
> | **Qwen0.5+Qwen0.6** | Wall-clock (h) | - | 0.15 | 0.29 | 0.44 | 0.87 | 5.59 |
> | | GPU Hours | - | 1.20 | 2.32 | 3.52 | 6.95 | 44.72 |
> | | MMLU Acc. (%) | 35.53 | 40.36 | **41.10** | **42.06** | **44.30** | **42.92** |
> | **Llama-1B+Qwen0.6** | Wall-clock (h) | - | 0.18 | 0.35 | 0.53 | 1.05 | 6.78 |
> | | GPU Hours | - | 1.44 | 2.80 | 4.24 | 8.44 | 54.24 |
> | | MMLU Acc. (%) | 35.53 | 28.66 | 41.16 | **43.98** | **44.37** | **44.42** |
> | **Qwen3-4B+Qwen0.6** | Wall-clock (h) | - | 0.16 | 0.32 | 0.48 | 0.96 | 6.17 |
> | | GPU Hours | - | 1.28 | 2.56 | 3.84 | 7.68 | 49.36 |
> | | MMLU Acc. (%) | 35.53 | 38.07 | 39.56 | 40.77 | **44.11** | **43.95** |
>
> The single GPU peak memory for C2C training on these three model pairs are 77.53GB, 79.16GB, and 78.67GB, respectively.
> As shown, accuracy increases rapidly in the first few hundred steps. We bold the values where C2C surpasses T2T on the MMLU benchmark.
> With just 300 steps ($<9$ GPU hours), C2C consistently outperforms T2T across all three model pairs and achieves performance comparable to the fully converged checkpoint.
> At inference time, C2C delivers $3.8 \times$, $1.5 \times$, $16.8 \times$ speedups over T2T on these model pairs. We believe this substantial efficiency gain and accuracy improvement fully justify the manageable one-time training overhead.

---

> ### Author Response · Authors · 2025-11-23
>
> ### Q2: Cost of C2C among different models
>
> > Is this idea reasonable? If you have two models of different architectures, transferring the KV cache does not seem like a good idea to me. The main reason is that the difference in the numeric value of KV are probably too high such that transforming one into another will be more difficult than generating the KV from text.
>
> We clarify that C2C is designed as a new communication paradigm to deliver richer information, rather than a pure efficiency optimization, like previous KV-Cache reuse methods.
> Thus, C2C does not aim to **replace** the Receiver's KV-Cache (which would indeed be difficult due to architectural differences), but to **enhance** it via residual semantic injection.
> Empirically, our experiments confirm that such enhancement is feasible across different architectures (e.g., Llama $\to$ Qwen), as shown in Tables 3 and 6.

---

### Official Review · Reviewer_Qkfp · 2025-10-29

**Soundness:** 3
**Presentation:** 2
**Contribution:** 3
**Rating:** 4
**Confidence:** 3

**Summary:**

This paper proposes a method, Cache-to-Cache (C2C), which enables two LLMs to communicate through the transfer of KV cache representations, bypassing the need to explicitly decode text. C2C projects and fuses the KV cache of a sharer model into a receiver model using a learned fuser that involves a projection, dynamic weighting, and learnable gating modules. The approach is transformable across model families and experiments show that receiver models in the C2C setup observe an improved accuracy and achieve a significant speedup.

**Strengths:**

1. The paper's primary strength is the novelty of it core idea, in using KV cache as a direct communication channel between two models, bypassing text generation. While KV cache sharing/re-use ideas exist for single-model inference, this paper proposes a novel latent communication between heterogeneous models.
2. The cache enrichment oracle is an interesting observation to decouple the effect of cache quality from cache length, demonstrating cache enrichment at fixed length improves quality.
3. The C2C trained MLP enables broad setups across model families and sizes.
4. Consistent accuracy and latency gains across models and tasks tested.

**Weaknesses:**

1. While the oracle experiments motivate the approach, there is limited analysis and justification for why KV-Cache is the optimal representation for inter-model communication, compared to other modes models may communicate with each other, such as through hidden-states.
2. Presentation quality needs to be improved. Some important design details such as alignment to enable heterogeneous communication are only briefly mentioned in main text. Figures and tables not being referenced also make it difficult to follow the paper (Figures not referenced in lines 205-206. Table 3 not referenced in the main text).
3. The performance benefits from C2C seem to diminish significantly when both the receiver model and sharer model size grow larger (Figure 6). A three layer MLP or the limited training may not be sufficient to capture the shift in distribution in KV cache representation for larger models.
4. The C2C fuser module must be trained for each specific sharer and receiver pair. This is a significant limitation that is not discussed. For $N$ distinct models in a multi-agent workflow, this implies training $O(N^2)$ fuser modules.
5. The paper's main evaluations are single-turn, single-answer benchmarks that do not require multi-agent collaboration. The paper would be stronger if C2C can be demonstrated to also work in tasks that involve models sharing responsibilities such as in planning or negotiation.

**Questions:**

1. The effective rank increases after fusion, but does this always correspond to "richer semantics"? Could you analyze what types of information are being transferred?
2. Table 4 shows performance gap between C2C and baseline approaches drop with longer contexts. Is this due to cache projection errors accumulating, or fundamental limitations of the approach?
3. What is the training time and GPU memory requirement for training a fuser? How does this compare to the potential inference savings?
4. Could you please clarify the latency calculation for C2C in Table 3? Does the reported C2C time (e.g., 0.40s for Qwen2.5-0.5B Sharer) include the time for both the Sharer's forward pass (0.34s) and the Receiver's forward pass (0.29s), plus the Fuser's computation? The C2C time seems lower than the sum of its parts.

---

> ### Author Response · Authors · 2025-11-23
>
> We sincerely thank Reviewer Qkfp for the constructive feedback.
> We appreciate your valuable recognition of the **novelty, observations from oracle experiments, broad applicability, and consistent accuracy and latency gains**.
> We address all the concerns and questions below.
>
> ### W1: Why use KV-Cache as the medium
>
> > While the oracle experiments motivate the approach, there is limited analysis and justification for why KV-Cache is the optimal representation for inter-model communication, compared to other modes models may communicate with each other, such as through hidden-states.
>
> We agree that hidden states may also be a good communication medium. We explore KV-Cache as a pioneering, feasible option for semantic communication beyond text. That being said, KV-Cache does have a few distinct advantages.
>
> One advantage is *system compatibility*. Since inference engines already store KV-Cache, C2C requires no additional memory structures to store intermediate results. Existing frameworks (e.g., SGLang [1], vLLM [2]) already provide sophisticated and efficient KV management, facilitating adoption with minimal interface changes.
>
> Additionally, LLMs are naturally trained to selectively store and extract contextual information through KV-Cache. As discussed in Equation 1, during decoding, hidden states primarily represent the current token, whereas the KV-Cache explicitly encodes the entire input and generation history.
>
> We sincerely thank the reviewer for the discussion and hope our work inspires further exploration of diverse multi-LLM communication media.
>
> [1] Zheng, Lianmin, et al. "Sglang: Efficient execution of structured language model programs." NeurIPS'24.
>
> [2] Kwon, Woosuk, et al. "Efficient memory management for large language model serving with pagedattention." Proceedings of the 29th symposium on operating systems principles. 2023.
>
> ### W2: Writing and Presentation
>
> > Presentation quality needs to be improved. Some important design details such as alignment to enable heterogeneous communication are only briefly mentioned in main text. Figures and tables not being referenced also make it difficult to follow the paper (Figures not referenced in lines 205-206. Table 3 not referenced in the main text).
>
> We sincerely thank the reviewer for the valuable suggestions. In the revised manuscript, we have:
>
> * Added more discussion on alignment strategies.
>
> * Refined the C2C paradigm description and formulation for greater clarity.
>
> * Ensured all Figures and Tables are properly referenced.
>
> * Proofread the entire text to improve presentation quality and flow.
>
> * Added additional key experiments and discussion from rebuttal.
>
> ### W3: C2C performance w.r.t model size scaling
>
> > The performance benefits from C2C seem to diminish significantly when both the receiver model and sharer model size grow larger (Figure 6). A three layer MLP or the limited training may not be sufficient to capture the shift in distribution in KV cache representation for larger models.
>
> We highlight that **C2C scales significantly better than the T2T baseline in Figure 6**.
> We attribute the diminishing gains to two factors:
>
> 1. Single-LLM accuracy already scales from 36% to 79% across 0.5B to 14B sizes, leaving the maximum accuracy improvement (margin towards 100% accuracy) already $3\times$ smaller for larger sizes.
>
> 2. We analyzed the Qwen-8B and Qwen-7B LLM prediction pattern on MMLU-Redux.
> Out of 5,632 total questions, Qwen3-8B answered 4,254 correctly, while Qwen2.5-7B answered 4,069 correctly, with an overlap of 3,624 questions answered correctly by both models.
> Questions that both models failed to answer remain inherently difficult. For example, when both models lack the required knowledge, neither T2T nor the proposed C2C can compensate for this deficiency. In such a scenario, the effect of LLM communication or collaboration might not be significant. Multi-LLM communication, regardless of medium, is fundamentally more effective in scenarios where the participating models possess complementary knowledge.

---

> ### Author Response · Authors · 2025-11-23
>
> ### W4: Pair-specific training
>
> > The C2C fuser module must be trained for each specific sharer and receiver pair. This is a significant limitation that is not discussed. For distinct models in a multi-agent workflow, this implies training O(N^2) fuser modules.
>
> We employ pair-specific training due to its manageable cost and strong downstream generalizability, which will be detailed in the response to Q3.
> Thus, we do not regard pair-specific training as a significant limitation in the current setting.
>
> As an exploration of a new paradigm, this work mainly focuses on communication between two models to set a foundation for future research.
> The communication via KV-Cache between multiple LLMs is a direction that requires future exploration, with different designs that can be employed.
> We appreciate the idea of training a universal fuser for all the model pairs, or reusing it for a model family. However, we think it is hard to implement as different models have diverse hidden dimensions that cannot fit in a single fuser.
> In response, we conduct preliminary explorations of alternative approaches that scale more efficiently with the number of Receivers and Sharers.
>
> **1. Scaling: Multiple Sharers, Single Receiver**
>
> We empirically find that fuser outputs are additive.
> Since they function as residual enhancements to the same Receiver KV, we can directly sum the outputs of multiple fusers, then add to the Receiver, similar to LoRA additivity [1].
>
> Formally, we are expanding Equation 3 to:
> $$ \mathcal{C}^{\mathcal{F}} = \\left\\{ \mathcal{C_n}(X) + \sum_{i=0}^{N-1} \mathcal{F_n^{\mathcal{S}i}}\left( \mathcal{C_n}(X) ,\mathcal{C_{\mathcal{G}(n)}^{\mathcal{S}i}}(X) \right) \\right\\}_{n=1}^{N} $$
>
> We tested this by fusing Qwen3-4B-Base (Sharer1) and Qwen2.5-1.5B-Math (Sharer2) into Qwen3-0.6B (Receiver). We use the original one-to-one trained fusers without any additional training for multiple-to-one.
> As shown below, the receiver successfully leverages both sharers to achieve higher performance than using either alone.
>
> |Type|MMLU Acc.|
> |:--|--:|
> |Receiver|35.53|
> |Sharer1|1.03|
> |Sharer2|39.86|
> |Sharer1 → Receiver|60.71|
> |Sharer2 → Receiver|46.13|
> |Sharer1, Sharer2 → Receiver|64.60|
>
> [1] Zhang, Jinghan, Junteng Liu, and Junxian He. "Composing parameter-efficient modules with arithmetic operation." NeurIPS'23.
>
> **2. Scaling: Multiple Sharers, Multiple Receivers.**
> We explore a path to use $O(N)$ fusers instead of $O(N^2)$ fusers to communicate among $N$ LLMs.
> The core idea is to project all Sharers into **one** shared *universal semantic space* ($U$) before fusing into Receivers.
> This splits the process into two stages, reducing complexity from $M \times N$ fusers to $M$ projectors + $N$ fusers.
>
> * Projection ($M$ Projectors): Each Sharer $s_i$'s cache is mapped to the latent space: $C_{s_i} \xrightarrow{P_{s_i}} U_{s_i}$.
>
> * Fusion ($N$ Fusers): Each Receiver $r_j$ fuses information from the shared space: $(U_{s_i}, C_{r_j}) \xrightarrow{F_{r_j}} C_{r_j}$.
>
> During training, each Receiver fuses information from $M$ projectors (or a sampled subset) to update its cache $C{r_j}$, then computes the standard SFT loss as in our paper.
> The total loss is the sum across all Receivers, used to update all Projectors and Fusers simultaneously.
> Once trained, it supports flexible C2C settings, including one-to-one, many-to-one, and one-to-many, enabling a full many-to-many landscape.
>
> We trained a prototype of this architecture using Qwen2.5-Coder-0.5B-Instruct and Qwen3-4B Base as Sharers, and Qwen3-0.6B and Qwen2.5-0.5B-Instruct as Receivers. As an initial exploration, we used the exact same hyperparameters as our pairwise experiments without specific tuning. The results below confirm the structural feasibility of this $O(N)$ approach:
>
> |Receiver|Receiver Acc.|C2C Acc. (coder-0.5b)|C2C Acc. (4b base)|C2C Acc. (both Sharers)|
> |:--|--:|--:|--:|--:|
> |**Qwen3-0.6B**|34.78|46.07|51.23|51.08|
> |**Qwen2.5-0.5B**|38.44|42.44|59.17|59.81|

---

> ### Author Response · Authors · 2025-11-23
>
> ### W5: Multi-agent benchmarks
>
> > The paper's main evaluations are single-turn, single-answer benchmarks that do not require multi-agent collaboration. The paper would be stronger if C2C can be demonstrated to also work in tasks that involve models sharing responsibilities, such as in planning or negotiation.
>
> We explore multi-agent collaboration by adopting a Math Problem Solving Agent Flow (Interpreter $\to$ Solver) from Cognify [1] on the GSM8K benchmark. The workflow consists of a problem interpreter that analyzes the math question and a problem solver that calculates the answer.
> For C2C, we use the prompt of math solver, so the problem interpretation is done by KV-Cache fusion. Here we present the result on GSM8K.
>
> |Method|Accuracy|
> |:--|--:|
> |Single Model (Qwen3-0.6B)|41.17|
> |T2T (Multi-agent flow)|61.18|
> |C2C|62.55|
> |T-C2C|78.01|
>
> As shown in the table, the multi-agent flow increase the accuracy of math problem solving by 20.01. C2C has an accuracy of 62.55, increasing the accuracy by 1.37 when compared with the multi-agent flow using T2T.
> The performance can be further elevated to 78.01 by using T-C2C, which let the math interpreter generate the interpretation and use C2C on both the question and the interpretation.
>
> ### Q1: Transferred information analysis
>
> > The effective rank increases after fusion, but does this always correspond to "richer semantics"? Could you analyze what types of information are being transferred?
>
> The effective rank measures the Shannon entropy of matrix eigenvalues and is widely used in PEFT methods as an indicator of generalization and robustness [1]. Therefore, we interpret the increase in effective rank as evidence that C2C fusion introduces additional informative and non-redundant signals to the Receiver.
>
> As this work enables LLM communication through an *implicit* way, it is challenging to *explicitly* define what types of information are being transferred. However, one possible form of transferred information is knowledge possessed by the Sharer but missing from the Receiver. Below is an example illustrating the knowledge transfer of models with different knowledge cutoffs.
>
> The knowledge cutoff for Qwen 2.5 is the end of 2023, so the model cannot answer questions related to things in 2024 correctly. Here is an example that C2C enables knowledge transfer:
>
> |**Prompt**| Who is the president of the United States in 2024?|
>
> |**Qwen2.5-7B**| As of my last update, the 2024 U.S. presidential election has not yet taken place, so I don't have information on who will be the president in 2024. |
>
> |**Qwen3-4b**| As of 2024, the President of the United States is **Joe Biden**. He has been in office since January 20, 2021, and is the 46th President of the United States. |
>
> |**C2C**| The president of the United States in 2024 is Joe Biden. He was inaugurated on January 20, 2021, and his term will end on January 20, 2025. |
>
> [1] Albert, Paul, et al. "Towards Higher Effective Rank in Parameter-Efficient Fine-tuning using Khatri-Rao Product." Proceedings of the IEEE/CVF International Conference on Computer Vision. 2025.
>
> ### Q2: Performance drop in longer contexts
>
> > Table 4 shows the performance gap between C2C and baseline approaches drop with longer contexts. Is this due to cache projection errors accumulating, or fundamental limitations of the approach?
>
> Previously, we followed Qwen3's recommended generation setup with sampling, which incurs randomness. Now, we change to greedy sampling to remove randomness and align with other experiments in the paper. We updated the result here and in the revised version of the paper.
>
> We test on LongBench with Qwen3-1.7B and Qwen2.5-1.5B, and the result does not show a drop in performance gap between C2C and baseline when increasing the context length.
> A possible reason that causes such a drop in the 0.6B and 0.5B model pair is that the Sharer is weak in the 8k+ setting, so it is hard for the weak Sharer to provide more useful contextual understanding compared with text, reducing the effect of C2C.
>
> |Model|0-4k (%)|4k-8k (%)|8k+ (%)|
> |:--|:--:|:--:|:--:|
> |Qwen3-0.6B|30.52|26.03|25.99|
> |Qwen2.5-0.5B-Instruct|24.94|23.18|16.44|
> |T2T_0.5_0.6|33.46|29.70|25.64|
> |C2C_0.5_0.6|37.31|34.01|30.72|
> |Delta (C2C - Receiver)|6.79|7.98|4.73|
>
> |Model|0-4k (%)|4k-8k (%)|8k+ (%)|
> |:--|:--:|:--:|:--:|
> |Qwen3-1.7B|42.57|37.47|36.90|
> |Qwen2.5-1.5B-Instruct|41.93|36.94|33.19|
> |T2T_1.5_1.7|43.68|36.62|34.57|
> |C2C_1.5_1.7|48.65|43.19|43.01|
> |Delta (C2C - Receiver)|6.08|5.72|6.11|

---

> ### Author Response · Authors · 2025-11-23
>
> ### Q3: C2C Training cost
>
> > What is the training time and GPU memory requirement for training a fuser? How does this compare to the potential inference savings?
>
> We employ pair-specific training due to its manageable cost and strong downstream generalizability. To quantify this, we present the training dynamics on 8 NVIDIA A800 GPUs in the table below.
>
> | Model Pair | Metric | 0 (Baseline) | 50 Steps | 100 Steps | 150 Steps | 300 Steps | Final (1929) |
> |:---|:---|:---:|:---:|:---:|:---:|:---:|:---:|
> | **Qwen0.5+Qwen0.6** | Wall-clock (h) | - | 0.15 | 0.29 | 0.44 | 0.87 | 5.59 |
> | | GPU Hours | - | 1.20 | 2.32 | 3.52 | 6.95 | 44.72 |
> | | MMLU Acc. (%) | 35.53 | 40.36 | **41.10** | **42.06** | **44.30** | **42.92** |
> | **Llama-1B+Qwen0.6** | Wall-clock (h) | - | 0.18 | 0.35 | 0.53 | 1.05 | 6.78 |
> | | GPU Hours | - | 1.44 | 2.80 | 4.24 | 8.44 | 54.24 |
> | | MMLU Acc. (%) | 35.53 | 28.66 | 41.16 | **43.98** | **44.37** | **44.42** |
> | **Qwen3-4B+Qwen0.6** | Wall-clock (h) | - | 0.16 | 0.32 | 0.48 | 0.96 | 6.17 |
> | | GPU Hours | - | 1.28 | 2.56 | 3.84 | 7.68 | 49.36 |
> | | MMLU Acc. (%) | 35.53 | 38.07 | 39.56 | 40.77 | **44.11** | **43.95** |
>
> The single GPU peak memory for C2C training on these three model pairs are 77.53GB, 79.16GB, and 78.67GB, respectively.
> As shown, accuracy increases rapidly in the first few hundred steps. We bold the values where C2C surpasses T2T on the MMLU benchmark.
> With just 300 steps ($<9$ GPU hours), C2C consistently outperforms T2T across all three model pairs and achieves performance comparable to the fully converged checkpoint.
> At inference time, C2C delivers $3.8 \times$, $1.5 \times$, $16.8 \times$ speedups over T2T on these model pairs. We believe this substantial efficiency gain and accuracy improvement fully justify the manageable one-time training overhead.
>
> ### Q4: Time breakdown
>
> > Could you please clarify the latency calculation for C2C in Table 3? Does the reported C2C time (e.g., 0.40s for Qwen2.5-0.5B Sharer) include the time for both the Sharer's forward pass (0.34s) and the Receiver's forward pass (0.29s), plus the fuser's computation? The C2C time seems lower than the sum of its parts.
>
> We clarify that the reported time includes both prefilling and decoding.
> C2C is faster than the sum of the individual runtimes of Sharer and Receiver because it skips the Sharer's decoding stage entirely, and only decodes once using the Receiver.
> The inference of C2C consists of four parts:
> (1) Sharer prefilling, (2) Receiver prefilling, (3) KV-Cache Fusion, and (4) Receiver decoding.
>
> We detail this breakdown below for the Qwen-0.6B (Receiver) + Qwen-0.5B (Sharer) pair:
>
> |Metric|Receiver|Sharer|T2T|C2C|
> |:--|:--:|:--:|:--:|:--:|
> |Input Token|170|187|103+332|170|
> |Output Token|11|19|10|12|
> |Communication Token|0|0|80|0|
> |-|-|-|-|-|
> |Prefill (ms)|27|20|21+32|47|
> |Decode (ms)|281|326|1312+231|308|
> |Fusion (ms)|0|0|0|90|
> |-|-|-|-|-|
> |Total Latency (ms)|308|346|1596|445|

---

### Official Review · Reviewer_TrrM · 2025-10-29

**Soundness:** 3
**Presentation:** 3
**Contribution:** 3
**Rating:** 4
**Confidence:** 3

**Summary:**

The paper proposes Cache-to-Cache (C2C), a way for one LLM to pass internal semantics to another by projecting and fusing their KV caches, instead of sending text messages. A small “cache projector” merges the sharer’s cache into the receiver’s cache, with a learned per-layer gate that decides where fusion helps. Two “oracle” studies motivate the design: cache enrichment without longer sequences, and cross-model cache transformability.

**Strengths:**

1) The two pilot studies are interesting and motivate the design: (a) cache enrichment at fixed cache length and (b) cross-model cache mapping that shows convertibility. They make the later choices like layer gating and fusion easier to trust.
2) Communicating through caches instead of text has clear novelty. It avoids extra decoding and can lower end-to-end latency while letting models share internal signals that plain text cannot carry.

**Weaknesses:**

1) Training cost vs benefit is not fully quantified. Please report GPU hours, peak memory, and wall time for training the fusion module per model pair, plus the speedup and accuracy gain at serving time. A clear trade-off table would help readers judge whether the training cost is paid back by latency savings and accuracy gains in real use.
2) Writing still need improvement. Some tables are not referenced in the paper.

**Questions:**

1) How does the method scale to more than two agents. For example, if there are N sharers and one receiver, do you fuse pairwise, do you use a learned mixer over all caches, or do you route by head or layer. What is the complexity and does interference grow with N.
2) Have you considered multi-agent benchmarks to test this setting. Examples include collaborative QA, team reasoning tasks, or tool-use suites where several models with different skills must coordinate. If not available, can you adapt existing agent benchmarks and report success rate, latency, and cost.

I would like to increase my score if the questions and weakness are addressed.

---

> ### Author Response · Authors · 2025-11-23
>
> We sincerely thank Reviewer TrrM for the positive feedback and constructive suggestions.
> We appreciate your valuable recognition of the **clear novelty, strong motivation from oracle experiments, and latency and performance benefits**.
> We address all the concerns and questions below.
>
>
> ### W1: C2C training cost
>
> > Training cost vs benefit is not fully quantified. Please report GPU hours, peak memory, and wall time for training the fusion module per model pair plus the speedup and accuracy gain at serving time.
>
> We employ pair-specific training due to its manageable cost and strong downstream generalizability. To quantify this, we present the training dynamics on 8 NVIDIA A800 GPUs in the table below.
>
> | Model Pair | Metric | 0 (Baseline) | 50 Steps | 100 Steps | 150 Steps | 300 Steps | Final (1929) |
> |:---|:---|:---:|:---:|:---:|:---:|:---:|:---:|
> | **Qwen0.5+Qwen0.6** | Wall-clock (h) | - | 0.15 | 0.29 | 0.44 | 0.87 | 5.59 |
> | | GPU Hours | - | 1.20 | 2.32 | 3.52 | 6.95 | 44.72 |
> | | MMLU Acc. (%) | 35.53 | 40.36 | **41.10** | **42.06** | **44.30** | **42.92** |
> | **Llama-1B+Qwen0.6** | Wall-clock (h) | - | 0.18 | 0.35 | 0.53 | 1.05 | 6.78 |
> | | GPU Hours | - | 1.44 | 2.80 | 4.24 | 8.44 | 54.24 |
> | | MMLU Acc. (%) | 35.53 | 28.66 | 41.16 | **43.98** | **44.37** | **44.42** |
> | **Qwen3-4B+Qwen0.6** | Wall-clock (h) | - | 0.16 | 0.32 | 0.48 | 0.96 | 6.17 |
> | | GPU Hours | - | 1.28 | 2.56 | 3.84 | 7.68 | 49.36 |
> | | MMLU Acc. (%) | 35.53 | 38.07 | 39.56 | 40.77 | **44.11** | **43.95** |
>
> The single GPU peak memory for C2C training on these three model pairs are 77.53GB, 79.16GB, and 78.67GB, respectively.
> As shown, accuracy increases rapidly in the first few hundred steps. We bold the values where C2C surpasses T2T on the MMLU benchmark.
> With just 300 steps ($<9$ GPU hours), C2C consistently outperforms T2T across all three model pairs and achieves performance comparable to the fully converged checkpoint.
> At inference time, C2C delivers $3.8 \times$, $1.5 \times$, $16.8 \times$ speedups over T2T on these model pairs. We believe this substantial efficiency gain and accuracy improvement fully justify the manageable one-time training overhead.
>
>
> ### W2: Writing and presentation
>
> > Writing still need improvement. Some tables are not referenced in the paper.
>
> We sincerely thank the reviewer for the valuable suggestions. In the revised manuscript, we have:
>
> * Added more discussion on alignment strategies.
>
> * Refined the C2C paradigm description and formulation for greater clarity.
>
> * Ensured all Figures and Tables are properly referenced.
>
> * Proofread the entire text to improve presentation quality and flow.
>
> * Added additional key experiments and discussion from rebuttal.

---

> ### Author Response · Authors · 2025-11-23
>
> ### Q1: Scaling: multiple sharers, single receiver
>
> > How does the method scale to more than two agents. For example, if there are N sharers and one receiver, do you fuse pairwise, do you use a learned mixer over all caches, or do you route by head or layer. What is the complexity and does interference grow with N.
>
> As an exploration of a new paradigm, this work mainly focuses on communication between two models to set a foundation for future research.
> The communication via KV-Cache between multiple LLMs is a direction that requires future exploration, with different designs that can be employed.
> In response, we conduct preliminary explorations of alternative approaches that scale more efficiently with the number of Receivers and Sharers.
>
> **1. Scaling: Multiple Sharers, Single Receiver**
>
> We empirically find that directly adding pair-wise fuser outputs is enough, without specific training or mixers.
> Since fuser outputs function as residual enhancements to the same Receiver KV, we can directly sum the outputs of multiple fusers, then add to the Receiver, similar to LoRA additivity [1].
>
> Formally, we are expanding Equation 3 to:
> $$ \mathcal{C}^{\mathcal{F}} = \\left\\{ \mathcal{C_n}(X) + \sum_{i=0}^{N-1} \mathcal{F_n^{\mathcal{S}i}}\left( \mathcal{C_n}(X) ,\mathcal{C_{\mathcal{G}(n)}^{\mathcal{S}i}}(X) \right) \\right\\}_{n=1}^{N} $$
>
> We tested this by fusing Qwen3-4B-Base (Sharer1) and Qwen2.5-1.5B-Math (Sharer2) into Qwen3-0.6B (Receiver). We use the original one-to-one trained fusers without any additional training for multiple-to-one.
> As shown below, the receiver successfully leverages both sharers to achieve higher performance than using either alone.
>
> |Type|MMLU Acc.|
> |:--|--:|
> |Receiver|35.53|
> |Sharer1|1.03|
> |Sharer2|39.86|
> |Sharer1 → Receiver|60.71|
> |Sharer2 → Receiver|46.13|
> |Sharer1, Sharer2 → Receiver|64.60|
>
> [1] Zhang, Jinghan, Junteng Liu, and Junxian He. "Composing parameter-efficient modules with arithmetic operation." NeurIPS'23.
>
> **2. Scaling: Multiple Sharers, Multiple Receivers.**
> We explore a path to use $O(N)$ fusers instead of $O(N^2)$ fusers to communicate among $N$ LLMs.
> The core idea is to project all Sharers into **one** shared *universal semantic space* ($U$) before fusing into Receivers.
> This splits the process into two stages, reducing complexity from $M \times N$ fusers to $M$ projectors + $N$ fusers.
>
> * Projection ($M$ Projectors): Each Sharer $s_i$'s cache is mapped to the latent space: $C_{s_i} \xrightarrow{P_{s_i}} U_{s_i}$.
>
> * Fusion ($N$ Fusers): Each Receiver $r_j$ fuses information from the shared space: $(U_{s_i}, C_{r_j}) \xrightarrow{F_{r_j}} C_{r_j}$.
>
> During training, each Receiver fuses information from $M$ projectors (or a sampled subset) to update its cache $C{r_j}$, then computes the standard SFT loss as in our paper.
> The total loss is the sum across all Receivers, used to update all Projectors and Fusers simultaneously.
> Once trained, it supports flexible C2C settings, including one-to-one, many-to-one, and one-to-many, enabling a full many-to-many landscape.
>
> We trained a prototype of this architecture using Qwen2.5-Coder-0.5B-Instruct and Qwen3-4B Base as Sharers, and Qwen3-0.6B and Qwen2.5-0.5B-Instruct as Receivers. As an initial exploration, we used the exact same hyperparameters as our pairwise experiments without specific tuning. The results below confirm the structural feasibility of this $O(N)$ approach:
>
> |Receiver|Receiver Acc.|C2C Acc. (coder-0.5b)|C2C Acc. (4b base)|C2C Acc. (both Sharers)|
> |:--|--:|--:|--:|--:|
> |**Qwen3-0.6B**|34.78|46.07|51.23|51.08|
> |**Qwen2.5-0.5B**|38.44|42.44|59.17|59.81|
>
> ### Q2: Multi-agent benchmarks
>
> > Have you considered multi-agent benchmarks to test this setting. Examples include collaborative QA, team reasoning tasks, or tool-use suites where several models with different skills must coordinate. If not available, can you adapt existing agent benchmarks and report success rate, latency, and cost.
>
> We explore multi-agent collaboration by adopting a Math Problem Solving Agent Flow (Interpreter $\to$ Solver) from Cognify [1] on the GSM8K benchmark. The workflow consists of a problem interpreter that analyzes the math question and a problem solver that calculates the answer.
> For C2C, we use the prompt of math solver, so the problem interpretation is done by KV-Cache fusion. Here we present the result on GSM8K.
>
> |Method|Accuracy|
> |:--|--:|
> |Single Model (Qwen3-0.6B)|41.17|
> |T2T (Multi-agent flow)|61.18|
> |C2C|62.55|
> |T-C2C|78.01|
>
> As shown in the table, the multi-agent flow increase the accuracy of math problem solving by 20.01. C2C has an accuracy of 62.55, increasing the accuracy by 1.37 when compared with the multi-agent flow using T2T.
> The performance can be further elevated to 78.01 by using T-C2C, which let the math interpreter generate the interpretation and use C2C on both the question and the interpretation.

---

### Official Review · Reviewer_JCWE · 2025-10-30

**Soundness:** 4
**Presentation:** 3
**Contribution:** 4
**Rating:** 8
**Confidence:** 3

**Summary:**

The paper proposes C2C, a novel paradigm for direct semantic communication between LLMs. Current multi-LLM systems communicate through a text-to-text (T2T) interface that requires re-encoding of conversation histories and therefore leads to loss in information. C2C proposes a way to let model exchange KV caches that encode richer contextual representations.

The key component of C2C is a cache fuser (a neural network) that projects the sharer model's KV-cache into the latent space of the receiver model. Then the fusing occurs through a dynamic weighting and gating mechanism to decide which layers are fused. C2C module can be trained using NTP with LLMs remain frozen.

Empirical result shows increase accuracy in various benchmark along with faster inference. Further behavior analysis shows increase in effective rank of the receiver's KV cache, verifying information gain.

**Strengths:**

1. Communication through KV-Cache is indeed a very interesting approach in improving communication between LLMs! KV-cache is known to be containing semantically rich, model-specific view of the context and this paper gives KV-cache a new role than being re-used for accelerating inference. And the motivation is well-justified through the oracle experiments.

2. The main experiments between homogeneous and heterogenous pairs of sharer and receivers showcase consistency: in almost all settings we see both improvement in accuracy compared to T2T and latency. I also like that the author conducts the effective rank analysis to show that there is indeed an information gain.

3. Clear ablation to justify every module: The design is based on findings from oracle experiment and each component seems to be adding performance gain gradually.

**Weaknesses:**

1. Pair-specific training and limited story on reuse: It seems that a retraining is required for every pair of sharer receiver models. I wonder how this additional cost scales up in more complex multi-LLM systems. There is likely cases where we would have more than two models and the communication between all of them are bidirectional. I also wonder how the cost of the C2C module changes as we have larger models it seems like the training cost is left undiscussed. In addition there is no experiment of showing how fusers can be "merged" (like training a single fuser by jointly optimize on a mixture of sharer and receivers)

2. Lack of failure mode analysis: Despite the accuracy gain, it seems unclear that what exact kind of question will fusion hurt, and how well this idea generalize to OOD settings. Right now this is justified through increased effective rank but some concrete examples would help for better understanding (similar to the <p> example but that is also slightly confusing in the way it's being presented right now).

3. Presentation needs to be improved: there seems to be unfinished section (for example missing figure in line 205-207)

**Questions:**

From weakness 1:

Q1: Is there a possibility for fuser reuse? Can you show, for example from the same family, that some of the fusers might be similar?

Q2: In addition is it possible to train a fuser for a pair of sharer group and receiver group?

From weakness 2:

Q3: Can you provide more granular breakdown of your evaluation to show which categories benefit/hurt using C2C versus T2T?

Q4: Some more qualitative example would be nice to have.

---

> ### Author Response · Authors · 2025-11-23
>
> We sincerely thank Reviewer JCWE for the positive feedback.
> We appreciate your valuable recognition of our **novelty, motivation, and validity**.
> We address all the concerns and questions below.
>
> ### W1: C2C training cost
>
> > Pair-specific training and limited story on reuse: It seems that a retraining is required for every pair of sharer receiver models.
>
> We employ pair-specific training due to its manageable cost and strong downstream generalizability. To quantify this, we present the training dynamics on 8 NVIDIA A800 GPUs in the table below.
>
> | Model Pair | Metric | 0 (Baseline) | 50 Steps | 100 Steps | 150 Steps | 300 Steps | Final (1929) |
> |:---|:---|:---:|:---:|:---:|:---:|:---:|:---:|
> | **Qwen0.5+Qwen0.6** | Wall-clock (h) | - | 0.15 | 0.29 | 0.44 | 0.87 | 5.59 |
> | | GPU Hours | - | 1.20 | 2.32 | 3.52 | 6.95 | 44.72 |
> | | MMLU Acc. (%) | 35.53 | 40.36 | **41.10** | **42.06** | **44.30** | **42.92** |
> | **Llama-1B+Qwen0.6** | Wall-clock (h) | - | 0.18 | 0.35 | 0.53 | 1.05 | 6.78 |
> | | GPU Hours | - | 1.44 | 2.80 | 4.24 | 8.44 | 54.24 |
> | | MMLU Acc. (%) | 35.53 | 28.66 | 41.16 | **43.98** | **44.37** | **44.42** |
> | **Qwen3-4B+Qwen0.6** | Wall-clock (h) | - | 0.16 | 0.32 | 0.48 | 0.96 | 6.17 |
> | | GPU Hours | - | 1.28 | 2.56 | 3.84 | 7.68 | 49.36 |
> | | MMLU Acc. (%) | 35.53 | 38.07 | 39.56 | 40.77 | **44.11** | **43.95** |
>
> As shown, accuracy increases rapidly in the first few hundred steps. We bold the values where C2C surpasses T2T on the MMLU benchmark.
> With just 300 steps ($<9$ GPU hours), C2C consistently outperforms T2T across all three model pairs and achieves performance comparable to the fully converged checkpoint.
> At inference time, C2C delivers $3.8 \times$, $1.5 \times$, $16.8 \times$ speedups over T2T on these model pairs. We believe this substantial efficiency gain and accuracy improvement fully justify the manageable one-time training overhead.
> But we agree that developing universal fusers to scale to larger numbers of LLMs is a promising direction for future work.
>
> ### Q1, Q2: Fuser reuse for #LLM scaling
>
> > Q1: Is there a possibility for fuser reuse? Can you show, for example from the same family, that some of the fusers might be similar?
> >
> > Q2: In addition is it possible to train a fuser for a pair of sharer group and receiver group?
>
> As an exploration of a new paradigm, this work mainly focuses on communication between two models to set a foundation for future research.
> The communication via KV-Cache between multiple LLMs is a direction that requires future exploration, with different designs that can be employed.
> We appreciate the idea of training a universal fuser for all the model pairs, or reusing it for a model family. However, we think it is hard to implement as different models have diverse hidden dimensions that cannot fit in a single fuser.
> In response, we conduct preliminary explorations of alternative approaches that scale more efficiently with the number of Receivers and Sharers.
>
>
> **1. Scaling: Multiple Sharers, Single Receiver**
>
> We empirically find that fuser outputs are additive.
> Since they function as residual enhancements to the same Receiver KV, we can directly sum the outputs of multiple fusers, then add to the Receiver, similar to LoRA additivity [1].
>
> Formally, we are expanding Equation 3 to:
> $$ \mathcal{C}^{\mathcal{F}} = \\left\\{ \mathcal{C_n}(X) + \sum_{i=0}^{N-1} \mathcal{F_n^{\mathcal{S}i}}\left( \mathcal{C_n}(X) ,\mathcal{C_{\mathcal{G}(n)}^{\mathcal{S}i}}(X) \right) \\right\\}_{n=1}^{N} $$
>
> We tested this by fusing Qwen3-4B-Base (Sharer1) and Qwen2.5-1.5B-Math (Sharer2) into Qwen3-0.6B (Receiver). We use the original one-to-one trained fusers without any additional training for multiple-to-one.
> As shown below, the receiver successfully leverages both sharers to achieve higher performance than using either alone.
>
> |Type|MMLU Acc.|
> |:--|--:|
> |Receiver|35.53|
> |Sharer1|1.03|
> |Sharer2|39.86|
> |Sharer1 → Receiver|60.71|
> |Sharer2 → Receiver|46.13|
> |Sharer1, Sharer2 → Receiver|64.60|
>
> [1] Zhang, Jinghan, Junteng Liu, and Junxian He. "Composing parameter-efficient modules with arithmetic operation." NeurIPS'23.

---

> ### Author Response · Authors · 2025-11-23
>
> **2. Scaling: Multiple Sharers, Multiple Receivers.**
> We explore a path to use $O(N)$ fusers instead of $O(N^2)$ fusers to communicate among $N$ LLMs.
> The core idea is to project all Sharers into **one** shared *universal semantic space* ($U$) before fusing into Receivers.
> This splits the process into two stages, reducing complexity from $M \times N$ fusers to $M$ projectors + $N$ fusers.
>
> * Projection ($M$ Projectors): Each Sharer $s_i$'s cache is mapped to the latent space: $C_{s_i} \xrightarrow{P_{s_i}} U_{s_i}$.
>
> * Fusion ($N$ Fusers): Each Receiver $r_j$ fuses information from the shared space: $(U_{s_i}, C_{r_j}) \xrightarrow{F_{r_j}} C_{r_j}$.
>
> During training, each Receiver fuses information from $M$ projectors (or a sampled subset) to update its cache $C{r_j}$, then computes the standard SFT loss as in our paper.
> The total loss is the sum across all Receivers, used to update all Projectors and Fusers simultaneously.
> Once trained, it supports flexible C2C settings, including one-to-one, many-to-one, and one-to-many, enabling a full many-to-many landscape.
>
> We trained a prototype of this architecture using Qwen2.5-Coder-0.5B-Instruct and Qwen3-4B-Base as Sharers, and Qwen3-0.6B and Qwen2.5-0.5B-Instruct as Receivers. As an initial exploration, we used the exact same hyperparameters as our pairwise experiments without specific tuning. The results below confirm the structural feasibility of this $O(N)$ approach:
>
> |Receiver|Receiver Acc.|C2C Acc. (coder-0.5b)|C2C Acc. (4b base)|C2C Acc. (both Sharers)|
> |:--|--:|--:|--:|--:|
> |**Qwen3-0.6B**|34.78|46.07|51.23|51.08|
> |**Qwen2.5-0.5B**|38.44|42.44|59.17|59.81|
>
>
> ### W2: Failure mode analysis
>
> > Lack of failure mode analysis: Despite the accuracy gain, it seems unclear that what exact kind of question will fusion hurt.
>
> We include an explicit failure mode analysis (Appendix) and a Limitations subsection.
> Our analysis shows that C2C performance degrades when *Sharer injects incorrect contextual understanding, misleading the Receiver*. This is most common when a very weak Sharer is paired with a strong Receiver.
>
> Note that this failure mode is inherent to multi-LLM collaboration, regardless of the communication medium, and is also observed in text-to-text communication.
> C2C mitigates this via the gating mechanism to suppress harmful Sharer inputs, though it is not fail-safe.
>
>
> ### Q3: Finer-grained performance analysis
>
> > Can you provide more granular breakdown of your evaluation to show which categories benefit/hurt using C2C versus T2T?
>
> To address the request for finer-grained performance analysis, we have added new figures and detailed results for different subcategories in Appendix A.3.3.
>
> For the **Qwen2.5-0.5B+Qwen3-0.6B** pair, C2C **outperforms T2T in 12 of 17 categories**. Notably, C2C yields substantial gains over T2T in categories like **History (+7.0%), Law (+7.6%), and Chemistry (+7.5%)**.
> Note that T2T retains a slight advantage in Math and Physics. We attribute this to the Sharer generating significantly longer CoT explanations in T2T for these subjects (avg. 222 tokens for Math vs. 80 overall), which conveys more explicit reasoning steps but incurs much higher latency.
>
> For the **Qwen3-4B+Qwen3-0.6B** pair, C2C **outperforms T2T on all 17 categories**. Using T2T in Engineering results in a 1% performance drop, illustrating a case where text communication from even a much stronger model fails to help the weak Receiver due to a lack of semantic transfer. In contrast, C2C successfully leverages the 4B model's contextual understanding to achieve a 24% increase in accuracy.
>
> ### W3: Writing and presentation
>
> > Presentation needs to be improved: there seems to be unfinished section (for example missing figure in line 205-207)
>
> We sincerely thank the reviewer for the valuable suggestions. In the revised manuscript, we have:
>
> * Added more discussion on alignment strategies.
>
> * Refined the C2C paradigm description and formulation for greater clarity.
>
> * Ensured all Figures and Tables are properly referenced.
>
> * Proofread the entire text to improve presentation quality and flow.
>
> * Added additional key experiments and discussion from rebuttal.

---

### Author Response · Authors · 2025-11-30
**Summary of Key Points**

We thank reviewers and AC for the valuable feedback and time. We appreciate the unanimous recognition of Cache-to-Cache (C2C) as **a new multi-LLM communication paradigm** beyond Text-to-Text (T2T). We have addressed all the concerns during the rebuttal. Below, we reiterate the key points:

---

### **Key recognitions:**

**1. Novel and interesting paradigm.** (`JCWE`, `TrrM`, `Qkfp`, `qYJq`)

C2C enables direct semantic communication between LLMs via KV-Cache instead of text. All reviewers consistently highlight its **clear novelty** and find it **very interesting**.

**2. Strongly motivated by oracle experiments.** (`JCWE`, `TrrM`, `Qkfp`).

Two oracle experiments provide **interesting observations** on the *benefit* and *convertibility* of KV-Cache, and **well-justify** the motivation and design choices of C2C.

**3. Consistent speedups and accuracy gains.** (`JCWE`, `Qkfp`, `Qkfp`, `qYJq`).

C2C works across **different model families, sizes, generations, specializations, and training stages**. It delivers **1.5-14.4× end-to-end speedup** and **3.1–5.4% accuracy gains** over text communication, supported by **clear ablation** on each component.

---

### **Key rebuttal additions:**

**1. Training cost and benefits.** (`JCWE` W1; `TrrM` W1; `Qkfp` Q3; `qYJq` Q1)

We detail the training costs of C2C fusers. With **$<9$ GPU hours** on A100-80GB GPUs, C2C already outperforms T2T across model pairs. This one-time general training enables **$2.5\times$ average inference speedup** across various downstream tasks, making training overhead clearly worthwhile.

**2. Scaling to multiple sharers, one receiver.** (`TrrM` Q1, `Qkfp` W4)

We show that C2C naturally extends from one-to-one to **multi-to-one communication**. Because fuser outputs act as residual enrichments, we can directly sum multiple fusers' outputs without any retraining. With two sharers, the receiver's MMLU accuracy improves **from 35.53% to 64.60%**, outperforming one-to-one results (46.13% and 60.71%).

**3. $O(N)$ scaling for $O(N^2)$ LLM pairs.** (`JCWE` Q1, Q2; `TrrM` Q1; `Qkfp` W4, )

We further explore an efficient design that uses **$O(N)$ fusers** instead of $O(N^2)$ pairwise fusers. It projects all sharers into one *universal semantic space* before fusing into receivers. The prototype shows that C2C's benefits still hold, supporting all-to-all communication among $N$ LLMs with only $O(N)$ fusers.

**4. Agentic math task evaluation.** (`TrrM` Q2, `Qkfp` W5)

We evaluate C2C under a practical multi-agent workflow: solving GSM8K with Math Problem Solving Agent (Interpreter $\to$ Solver). C2C achieves 62.55% accuracy, surpassing the T2T agent baseline (61.18%). Combining T2T with C2C further boosts accuracy to **78.01% (+16.83%)**.

**Additional clarifications.**

We also provide fine-grained performance (`JCWE` W2, Q3, `Qkfp` Q2) and information (`Qkfp` Q1) analysis, communication medium discussion (`Qkfp` W1), size-scaling discussion (`Qkfp` W3, `qYJq` W1), runtime breakdown (`Qkfp` Q4; `qYJq` W2, Q2), and improve writing details (`JCWE` W3; `TrrM` W2; `Qkfp` W2). The paper has been revised accordingly.

---

### Meta-Review · Area_Chair_dtvJ · 2026-01-07

**Summary:**

The paper studies adapting the KV cache of the prompt on one model ("source") to another model ("receiver"), notated in eq (4). The adapter is another module mapping between the KV caches learned via a standard cross-entropy loss on the receiver. Experimentally the results in Table 3 show positive transfer into Qwen3 0.6B and the other ablations show some key insights the design decisions. While there are some minor concerns discussed in the next section the authors should finish addressing for the final version, the paper clearly demonstrates the key idea and will be interesting for the community.

One minor consideration for the final presentation, I find the term "communication" slightly incorrect to describe the adaptation that is happening here: the source nor receiver models are never updated during the training process, so they aren't "communicating" in the sense that neither the sender nor receiver know they are sending or receiving the prompt's KV cache items. I think "adapter" is a more precise and accurate term for the transfer being shown in the paper.

**Reviewer Concerns:**

The reviewers raised some concerns around the following points:

+ Diminishing benefit with larger sharer and receiver models
+ Training requirement for each specific sharer and receiver pair
+ Training cost analysis
+ Scaling with multiple sharers
+ Presentation

The rebuttal mostly addresses these and they are important to discuss in the final version. I do not see them as holding back the paper from a publication.

I do raise one last concern with the paper on the weak baselines in the paper and recommend the authors to take this into consideration for the final version of the paper: C2C benefits tremendously from using 500k samples from OpenHermes and the other baselines considered don't fine-tune on the same data. So it's not clear if the gain is coming from adapting the KV cache or fine-tuning on another dataset. For example, maybe even fine-tuning the receiver model on OpenHermes 2.5 directly could increase the performance. Also, the T2T baseline seems extremely weak, simply prompting "In one clear sentence, describe the most essential background knowledge needed to answer the question": more sophisticated prompting strategies could be used to better-transfer knowledge between models.

Despite these concerns, the idea of transferring between the KV caches of different models is novel and the paper does clearly demonstrate a gain from it, so I still recommend an accept.

**Reviewer Scores:**

The would have been mostly consistent, possibly with weak rejects raising

---

### Decision · Program_Chairs · 2026-01-26

Accept (Poster)